# Probe Pruning: Accelerating LLMs through Dynamic Pruning via Model-Probing

**Qi Le**[1], **Enmao Diao**, **Ziyan Wang**[2], **Xinran Wang**[1], **Jie Ding**[1], **Li Yang**[2], **Ali Anwar**[1]
[1]University of Minnesota    [2]University of North Carolina at Charlotte
`{le000288, wang8740, dingj, aanwar}@umn.edu,`
`diao_em@hotmail.com,{zwang53, lyang50}@charlotte.edu`

## Abstract

We introduce Probe Pruning (PP), a novel framework for online, dynamic, structured pruning of Large Language Models (LLMs) applied in a batch-wise manner. PP leverages the insight that not all samples and tokens contribute equally to the model's output, and probing a small portion of each batch effectively identifies crucial weights, enabling tailored dynamic pruning for different batches. It comprises three main stages: probing, history-informed pruning, and full inference. In the probing stage, PP selects a *small yet crucial* set of hidden states, based on residual importance, to run a few model layers ahead. During the history-informed pruning stage, PP strategically integrates the probing states with historical states. Subsequently, it structurally prunes weights based on the integrated states and the PP importance score, a metric developed specifically to assess the importance of each weight channel in maintaining performance. In the final stage, full inference is conducted on the remaining weights. A major advantage of PP is its compatibility with existing models, as it operates *without requiring additional neural network modules or fine-tuning*. Comprehensive evaluations of PP on LLaMA-2/3 and OPT models reveal that even minimal probing-using just 1.5% of FLOPs-can substantially enhance the efficiency of structured pruning of LLMs. For instance, when evaluated on LLaMA-2-7B with WikiText2, PP achieves a $2.56\times$ lower ratio of performance degradation per unit of runtime reduction compared to the state-of-the-art method at a 40% pruning ratio. Our code is available at `https://github.com/Qi-Le1/Probe_Pruning`.

## 1 Introduction

Large Language Models (LLMs) (Vaswani et al., 2017; Zhang et al., 2022; Touvron et al., 2023; Diao et al., 2024) have recently achieved significant success, leading to the development of numerous applications (OpenAI, 2023; Anand et al., 2023). However, the inference for these models, often containing billions of parameters, presents challenges. These challenges primarily arise from the substantial computational demands and the risk of high latency (Ma et al., 2023).

Structured pruning is a promising hardware-friendly approach to reduce computational consumption and accelerate inference (Yuan et al., 2021). It removes complete structures from models, such as weight channels and attention heads. Compared with other methods like unstructured pruning (Frantar & Alistarh, 2023; Sun et al., 2023), parameter sharing (Diao et al., 2019), offloading (Rasley et al., 2020; Diao et al., 2022; 2024), and quantization (Dettmers et al., 2022; Lin et al., 2023; Frantar et al., 2022), structured pruning reduces computational resources and speeds up inference without requiring specific hardware. However, when applied to LLMs, structured pruning often results in a performance gap compared to dense models (Wang et al., 2024b).

A major factor contributing to the performance gap in LLMs may be the emergence of significant outlier phenomena in internal representations (Dettmers et al., 2022; Liu et al., 2024; Sun et al., 2024). Current advanced structured pruning methods typically utilize calibration datasets to assess the importance of weights using pruning metrics. For example, the FLAP method (An et al., 2024) uses a calibration dataset to compute fluctuation metrics for each input feature and its corresponding channel in attention or MLP block weight matrices, specifically in the output projection (O) or fully connected layer 2 (FC2). Similarly, LLM-Pruner (Ma et al., 2023) employs approximated second-order Taylor expansions of the error function, calculated using a calibration dataset, to eliminate the

least important coupled structures. Although the calibration dataset provides valuable insights for pruning by identifying non-critical weights, this approach overlooks the batch-dependent nature of outlier properties in LLMs (Liu et al., 2023b; Song et al., 2023; Liu et al., 2024; Sun et al., 2024), which vary across different input batches and cannot be accurately predicted prior to inference. Experimental illustrations can be found in Appendix B.3. Consequently, pruning decisions based solely on calibration dataset may fail to address these dynamic outliers during real-time inference, resulting in suboptimal model performance. Fine-tuning can serve as a method to recover model performance (Wang et al., 2024b), but it is resource-intensive and may be impractical for certain real-world applications.

To effectively handle batch-dependent outliers and reduce the performance gap between pruned and dense models without extensive fine-tuning, we propose Probe Pruning (PP). PP is an online dynamic structured pruning framework that prunes the model during inference based on each batch's hidden states. Notably, PP relies solely on the original model structure and hidden states, *without requiring additional neural network modules or fine-tuning*. We overcome two key challenges:

- **Leveraging Calibration Dataset may Introduce Biases**: Relying exclusively on the calibration dataset may introduce biases, as the pruned channels are entirely determined by the calibration dataset. For example, when FLAP used the WikiText2 validation set as a calibration dataset, it achieved a perplexity of 18.5 on the WikiText2 test set of LLaMA-2-7B with a 40% pruning ratio. In contrast, using the C4 dataset as a calibration dataset, the perplexity increased to 38.9 on the WikiText2 test set. *We propose history-informed pruning with importance-scaled fusion to leverage the benefits of the calibration dataset while minimizing associated biases.*

- **Dynamic Pruning Without Access to Intermediate Hidden States**: Deciding online which channels to prune during inference for each batch is challenging. Without gradients, calculating pruning metrics for attention and MLP blocks requires intermediate hidden states, which are the input tensors to the attention output projection and MLP's FC2 layer. These states are unavailable when the input hidden states initially enter these blocks. Moreover, not all samples and tokens contribute equally to the model's output, and large-magnitude outliers in LLMs often have a significant impact on the model's behavior. *Therefore, we propose a probing method that selects key samples and tokens from the input hidden states, runs a few model layers ahead, and obtains intermediate hidden state information.* Without such probing, accessing intermediate hidden states requires significant computational costs.

Specifically, PP leverages a *small yet crucial* segment of hidden states to run a few model layers ahead and capture the probe's intermediate hidden states, which contain essential information for guiding pruning decisions within the attention or MLP blocks of the current batch. By strategically integrating the probing states with historical states, we can dynamically determine which channels to prune. After pruning the weight channels, we run full inference on the remaining weights. Furthermore, our probing is minimal yet effective: for example, it operates with only 5% of the samples and 50% of the tokens, utilizing just 1.5% of the floating point operations (FLOPs) of dense model inference, and yet it has proven effective. Experimental results confirm that this minimal probe effectively captures critical intermediate hidden state information.

## 2 RELATED WORK

**Pruning with Calibration Dataset.** Pruning methods can be broadly classified into two categories (Yuan et al., 2021): *unstructured pruning* and *structured pruning*. Unstructured pruning (LeCun et al., 1989; Hassibi et al., 1993; Han et al., 2015; Mushtaq et al., 2021; Li et al., 2021; Soltani et al., 2021; Yang et al., 2022; Diao et al., 2023; Liu et al., 2023a; Li et al., 2024b;a; Dong et al., 2024) removes individual weights, whereas structured pruning (Li et al., 2016; Liu et al., 2017; He et al., 2019; Diao et al., 2020; Fang et al., 2023) removes complete structures from the model, such as channels and attention heads. Pruning LLMs often involves calibration datasets due to the emergence of outliers in their internal representations. For unstructured pruning, SparseGPT (Frantar & Alistarh, 2023) uses synchronized second-order Hessian updates to solve row-wise weight reconstruction problems and update weights. Wanda (Sun et al., 2023) introduces a pruning metric that considers both the magnitude of weights and activation values to determine which weights to prune. For structured pruning, FLAP (An et al., 2024) introduces a fluctuation metric to decide which weight channels to prune. LLM-Pruner (Ma et al., 2023) employs approximated second-order Taylor expansions of the

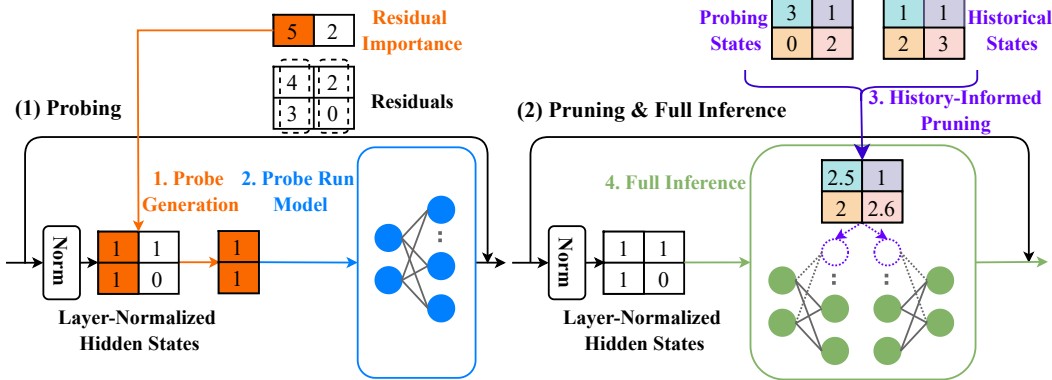

Figure 1: Probe Pruning (PP) is executed in four stages: (1) PP selects key samples and tokens from the layer-normalized hidden states, based on residual importance, to create a *small yet crucial* probe. (2) PP deploys this probe to run a few model layers ahead and obtains the probe's intermediate hidden states. (3) PP integrates the probing states with historical states and uses the integrated states to calculate the pruning metric and prune weight channels. (4) PP performs full inference on the remaining weights.

error function to remove the least important coupled structures and then applies fine-tuning to recover model performance. LoRA-Prune (Zhang et al., 2023) uses a LoRA (Hu et al., 2021)-guided pruning metric that leverages the weights and gradients of LoRA to direct the iterative process of pruning and tuning. However, the fine-tuning process requires substantial computational resources (Hoffmann et al., 2022), and we have found that fine-tuning might cause LLMs to lose their generalizability; for example, they may perform worse on certain downstream tasks, such as commonsense reasoning tasks.

**Large-Magnitude Outliers of LLMs.** Unlike small neural networks, LLMs exhibit large-magnitude outlier features (Kovaleva et al., 2021; Dettmers et al., 2022; 2023; Schaeffer et al., 2024; Sun et al., 2024). Dettmers et al. (2022) shows that these large-magnitude features begin to emerge when the size of LLMs exceeds 6.7 billion parameters, and these outlier features are concentrated in certain channels. The phenomenon of massive activations (Sun et al., 2024; Liu et al., 2024) has been observed, where a few activations exhibit significantly larger values than others, potentially leading to the concentration of attention probabilities on their corresponding tokens. These emergent properties suggest the need to customize the pruning of channels for different batches to maintain model performance. This observation motivates us to propose Probe Pruning.

## 3  NOTATIONS AND PRELIMINARIES

An LLM $\mathcal{M}$ consists of $L$ blocks, each of which can be either an attention block or a Multi-Layer Perceptron (MLP) block. Each attention block is characterized by four linear projections: Query (Q), Key (K), Value (V), and Output (O). Similarly, each MLP block includes two linear layers: Fully Connected layer 1 (FC1) and Fully Connected layer 2 (FC2).

Each block $l$ transforms the input hidden state $\mathbf{X}^l \in \mathbb{R}^{N \times S \times D}$ into the output hidden state $\mathbf{X}^{l+1} \in \mathbb{R}^{N \times S \times D}$. Here, $N$, $S$, and $D$ denote the batch size, sequence length, and feature dimension, respectively. The transformations in each block $l$ can be expressed as:

$$\mathbf{X}^{l+1} = \mathbf{X}^l + \mathcal{F}^l(\mathbf{X}^l), \tag{1}$$

where $\mathcal{F}^l$ encompasses all transformations within block $l$. This function can be further decomposed as:

$$\mathcal{F}^l(\mathbf{X}^l) = \mathbf{X}^{l,\text{int}}(\mathbf{W}^{l,\text{final}})^T, \quad \mathbf{X}^{l,\text{int}} = \mathcal{T}^l(\text{LN}^l(\mathbf{X}^l)), \tag{2}$$

where $\mathcal{T}^l$ represents all intermediate transformations applied to the input hidden state $\mathbf{X}^l$, excluding the layer normalization $\text{LN}^l$ and final weight matrix $\mathbf{W}^{l,\text{final}} \in \mathbb{R}^{C_\text{out} \times C_\text{in}}$. The final weight matrix is either the Output projection (O) in an attention block or FC2 in an MLP block. The intermediate hidden state $\mathbf{X}^{l,\text{int}} \in \mathbb{R}^{N \times S \times C_\text{in}}$ results from applying these intermediate transformations to $\mathbf{X}^l$. Additionally, we define the *residual importance* as the $L^2$ norm of the input hidden states $\mathbf{X}^l$ across specific dimensions, a concept further detailed in Section 4.2.

In structured pruning of LLMs, entire coupled structures are pruned (Ma et al., 2023; An et al., 2024). Specifically, in block $l$, the preceding weight matrices are adjusted by pruning their output channels, which correspond one-to-one with the input channels pruned by the final weight matrix. For example, in an MLP block, the weight matrices are adjusted based on the set of unpruned channel indices $\mathbb{C}^l \subseteq \{1, 2, \ldots, C_{\text{in}}\}$ as follows:

$$\tilde{\boldsymbol{W}}^{l,\text{FC1}} = \boldsymbol{W}^{l,\text{FC1}}[\mathbb{C}^l, :], \quad \tilde{\boldsymbol{W}}^{l,\text{FC2}} = \boldsymbol{W}^{l,\text{FC2}}[:, \mathbb{C}^l], \tag{3}$$

where $\tilde{\boldsymbol{W}}^{l,\text{FC1}} \in \mathbb{R}^{|\mathbb{C}^l| \times C_{\text{in}}}$ and $\tilde{\boldsymbol{W}}^{l,\text{FC2}} \in \mathbb{R}^{C_{\text{out}} \times |\mathbb{C}^l|}$. The notation $|\mathbb{C}^l|$ represents the cardinality of $\mathbb{C}^l$. Similarly, in the attention block, attention heads can be treated as coupled structures (Ma et al., 2023; An et al., 2024), and entire attention heads are pruned.

## 4 METHODOLOGY

The objective of Probe Pruning (PP) is to implement online dynamic structured pruning in a batch-wise manner. The main idea of our work is illustrated in Figure 1. Our core strategy involves: (1) **Probing** (Sections 4.1 and 4.2), which consists of two steps: first, generating a probe based on residual importance; second, using the probe to run the unpruned model to gather valuable intermediate hidden state information. (2) **History-informed pruning** (Section 4.3), which carefully merges the probing states with historical states using importance-scaled fusion to capture the essential characteristics of each batch. Afterward, we prune the model using a novel pruning metric (Section 4.4) that more effectively selects channels for pruning than existing metrics.

### 4.1 PROBING

We introduce a novel concept called probing, which leverages the existing model structure and hidden states to form a predictive mechanism. Specifically, when the input hidden states reach block $l$, probing first utilizes residual importance to select key samples and tokens, forming the probe $\mathbf{P}^l$ from $\text{LN}^l(\mathbf{X}^l)$. $\text{LN}^l$ represents the layer normalization at block $l$. The process of probe generation is detailed in the next section. It then runs the intermediate transformation in block $l$, denoted by $\mathcal{T}^l(\mathbf{P}^l)$. Notably, effective probing consumes few computational resources and can obtain important intermediate-state information to guide pruning decisions.

**Upper Bound of Probing.** As an alternative, we can generate the probe by using all the input hidden states in the current batch, $\mathbf{P}^l = \text{LN}^l(\mathbf{X}^l)$, a method we refer to as *Full-Batch Probing*. By utilizing the entire batch without reducing the dimensions $N$ or $S$, Full-Batch Probing captures the complete intermediate hidden state information, which could potentially lead to optimal pruning performance. However, this approach significantly increases computational resource requirements and latency. Therefore, Full-Batch Probing serves as a theoretical upper bound for our method. Our aim for PP is to select pruning channels similar to those chosen by Full-Batch Probing. We believe that a higher proportion of common pruning channels between PP and Full-Batch Probing indicates better model performance and higher quality of the probe.

**Why Does Probing Work?** Probing is effective because not all samples and tokens contribute equally to the model's output, and large-magnitude outliers in LLMs significantly influence the model's behavior. In natural language sequences, certain tokens carry more semantic or syntactic significance than others (Xiao et al., 2023; Sun et al., 2024; Liu et al., 2024). By selecting key samples and tokens based on residual importance, the probe focuses on the most informative parts within the batch. This targeted approach allows the probe to capture essential intermediate hidden state information that is most influential in determining which channels can be pruned. Consequently, even though the probe processes a reduced subset of the batch, it provides sufficient insight to guide pruning decisions, potentially achieving results comparable to Full-Batch Probing with significantly lower computational costs.

**Computational Complexity.** Only minimal computational complexity is required for probing. Specifically, for an LLM characterized by six linear transformations per attention and MLP block (Q/K/V/O and FC1/FC2) that incorporate weight transformations and the attention mechanism, the dense matrix computational complexity for an LLM totals $O(6NSC_{\text{in}}C_{\text{out}} + 2NS^2C_{\text{in}})$. For probing, by reducing the batch size to $x\%$ and the sequence length to $y\%$ of their original sizes, the complexity reduces to $O(4x\% \cdot y\% \cdot NSC_{\text{in}}C_{\text{out}} + 2x\% \cdot (y\%)^2 \cdot NS^2C_{\text{in}})$.

### 4.2 PROBE GENERATION

PP measures the *residual importance*, which is the $L^2$ norm of $\mathbf{X}^l$ across specific dimensions to identify key samples and tokens. Once identified, these key samples and tokens are selected from

---

**Algorithm 1:** Probe Pruning

---

**Input:** An LLM $\mathcal{M}$ with $L$ blocks, each containing the Transformation $\mathcal{F}^l$, the Intermediate
transformation $\mathcal{T}^l$, and Layer Normalization $\text{LN}^l$; calibration dataset $D$; Inference batches $B$.
**System executes:**
   Run the calibration dataset $D$ using model $\mathcal{M}$ to obtain historical states $\mathbf{V}$.
   **for** $t$-th batch $B^t$ **do**
      Initialize the hidden state $\mathbf{X}^0$ for batch $B^t$.
      **for** each block $l = 0, \ldots, L-1$ **do**
         Generate a probe $\mathbf{P}^l$ from $\text{LN}^l(\mathbf{X}^l)$, utilizing the residual importance (Section 4.2).
         Use $\mathbf{P}^l$ to execute the intermediate transformation of block $l$ and gather the resulting
           intermediate hidden states, denoted as $\mathbf{X}^{l,\text{int,probe}} = \mathcal{T}^l(\mathbf{P}^l)$.
         Use importance-scaled fusion to integrate the probing states $\mathbf{X}'^{l,\text{int,probe}}$ with historical states
           (Section 4.3).
         Compute the PPsp pruning metric from the integrated states (Section 4.4), and subsequently
           prune the weight channels accordingly.
         Execute full inference on $\mathbf{X}^l$ using the pruned weights $\tilde{W}^l$, denoted by $\tilde{\mathcal{F}}^l(\mathbf{X}^l)$.
      **end**
   **end**

---

$\text{LN}^l(\mathbf{X}^l)$ to generate a probe for block $l$, where $\text{LN}^l$ denotes layer normalization at block $l$. We do not utilize the importance derived from $\text{LN}^l(\mathbf{X}^l)$ to identify key samples and tokens because layer normalization substantially alters the input hidden states.

To measure the residual importance along the target dimension, we compute the $L^2$ norm of $\mathbf{X}^l$ across non-target dimensions. The target dimension may be either the batch or sequence dimension.

$$\mathbf{U}_i^{l,\text{batch}} = \|\mathbf{X}_{i,:,:}^l\|_2, \quad \text{for } i = 1, \ldots, N, \tag{4}$$

$$\mathbf{U}_j^{l,\text{seq}} = \|\mathbf{X}_{:,j,:}^l\|_2, \quad \text{for } j = 1, \ldots, S. \tag{5}$$

After computing the importance scores, we sort them in descending order and store the indices in $\mathbb{I}$:

$$\mathbb{I}^{l,\text{batch}} = \text{argsort}(-\mathbf{U}^{l,\text{batch}}), \tag{6}$$

$$\mathbb{I}^{l,\text{seq}} = \text{argsort}(-\mathbf{U}^{l,\text{seq}}). \tag{7}$$

Using the sorted indices, we then generate the probe by selecting the top $x\%$ of samples or $y\%$ of tokens from the layer-normalized $\mathbf{X}^l$:

$$\mathbf{P}^l = \begin{cases} \text{LN}^l(\mathbf{X}^l)_{\mathbb{I}_{:x\%}^{l,\text{batch}},:,:} & \text{if selecting top } x\% \text{ of samples,} \\ \text{LN}^l(\mathbf{X}^l)_{:,\mathbb{I}_{:y\%}^{l,\text{seq}},:} & \text{if selecting top } y\% \text{ of tokens.} \end{cases} \tag{8}$$

This method ensures that the probe consists of the most significant samples and tokens, as ranked by their importance scores.

PP implements a sequential approach to prune both sequence and batch dimensions effectively. Initially, the top $y\%$ of tokens are selected from the residual $\mathbf{X}^l$, guided by Eqs. (5) and (7), leveraging the sequence distribution within the current batch: $\mathbf{X}_{:,\mathbb{I}_{:y\%}^{l,\text{seq}},:}^l$. Subsequently, we apply this reduced sequence set to determine the top $x\%$ of samples using Eqs. (4) and (6), resulting in the indices $\mathbb{I}^{l,\text{batch}|\text{seq}}$. Finally, we select the key samples and tokens for probe generation as $\text{LN}^l(\mathbf{X}^l)_{\mathbb{I}_{:x\%}^{l,\text{batch}|\text{seq}},\mathbb{I}_{:y\%}^{l,\text{seq}},:}$.

## 4.3 HISTORY-INFORMED PRUNING WITH IMPORTANCE-SCALED FUSION

The intermediate hidden states of the probe, given by

$$\mathbf{X}^{l,\text{int,probe}} = \mathcal{T}^l(\mathbf{P}^l) \tag{9}$$

contain crucial information that guides pruning decisions. However, when the probe is very small—for instance, when $N$ and $S$ are reduced to 5%—they might lead to inappropriate pruning decisions due

to limited context. To address this issue and enhance performance, we introduce history-informed pruning with importance-scaled fusion.

To simplify notation, we omit the superscript $l$, which denotes the block number, in this section. For intermediate hidden states $\mathbf{X}^{\text{int}}$ of shape $(N, S, C_{\text{in}})$, the following relationship holds:

$$\sum_{j=1}^{S} \sum_{i=1}^{N} (\mathbf{X}_{i,j,k}^{\text{int}})^2 = \sum_{j=1}^{S} ||\mathbf{X}_{:,j,k}^{\text{int}}||_2^2 = ||\mathbf{X}_{:,:,k}^{\text{int}}||_2^2 \tag{10}$$

We compress the batch dimension in the first step of Eq. 10 to store historical states because memory limitations prevent storing the intermediate hidden states of all samples. We sum over the sequence dimension in the second step of Eq. 10 to obtain the tensor in shape $\mathbb{R}^{C_{\text{in}}}$, which is used to compute the pruning metric (see Section 4.4).

As in previous studies (Sun et al., 2023; An et al., 2024), we process the calibration dataset $D$ using the model $\mathcal{M}$ to obtain initial historical states. For each block, initial historical states are represented by $\mathbf{V}|^0 \in \mathbb{R}^{S \times C_{\text{in}}}$, computed as the first step of Eq. 10 to reduce the batch dimension: $\mathbf{V}|^0 = ||\mathbf{X}_{:,j,k}^{\text{int}}||_2^2 = \sum_{i=1}^{N} (\mathbf{X}_{i,j,k}^{\text{int}})^2$. Similarly, to reduce the batch dimension of probe's intermediate hidden states $\mathbf{X}^{\text{int,probe}} \in \mathbb{R}^{x\% \cdot N \times y\% \cdot S \times C_{\text{in}}}$, we calculate probing states as $||\mathbf{X}_{:,j,k}^{\text{int,probe}}||_2^2 = \sum_{i=1}^{x\% \cdot N} (\mathbf{X}_{i,j,k}^{\text{int,probe}})^2$.

**Importance-Scaled Fusion.** Since probing can be performed with selected tokens, it is necessary to align the sequence dimension. We define $\mathbf{V}^{\text{probe}} = \mathbf{V}_{\mathbb{I}_{:,y\%}^{\text{seq}},:}$, where $\mathbf{V}^{\text{probe}} \in \mathbb{R}^{y\% \cdot S \times C_{\text{in}}}$ and $\mathbb{I}_{:,y\%}^{\text{seq}}$, obtained from Eq. 7, represents the indices of the top $y\%$ of tokens. We then apply importance-scaled fusion to obtain integrated states:

$$\hat{\mathbf{X}}^{\text{int,probe}} = \frac{||\mathbf{X}_{:,j,k}^{\text{int,probe}}||_2^2}{||\mathbf{X}_{:,j,k}^{\text{int,probe}}||_2^2 + \mathbf{V}^{\text{probe}}} \cdot ||\mathbf{X}_{:,j,k}^{\text{int,probe}}||_2^2 + \frac{\mathbf{V}^{\text{probe}}}{||\mathbf{X}_{:,j,k}^{\text{int,probe}}||_2^2 + \mathbf{V}^{\text{probe}}} \cdot \mathbf{V}^{\text{probe}}, \tag{11}$$

where $\hat{\mathbf{X}}^{\text{int,probe}} \in \mathbb{R}^{y\% \cdot S \times C_{\text{in}}}$. Following the second step of Eq. 10, we sum $\hat{\mathbf{X}}^{\text{int,probe}}$ over the sequence dimension to obtain $\sum_{j=1}^{y\% \cdot S} \hat{\mathbf{X}}_{j,k}^{\text{int,probe}}$. Note that without importance-scaled fusion, $\sum_{j=1}^{y\% \cdot S} \hat{\mathbf{X}}_{j,k}^{\text{int,probe}}$ can reduce to $||\mathbf{X}_{:,:,k}^{\text{int}}||_2^2$. Then, we use $\mathbf{W}^{\text{final}}$ and $\sum_{j=1}^{y\% \cdot S} \hat{\mathbf{X}}_{j,k}^{\text{int,probe}}$ as a surrogate of $||\mathbf{X}_{:,:,k}^{\text{int}}||_2^2$ to calculate the pruning metric based on Eq. (15), and prune the weight channels accordingly. Finally, we run full inference on the remaining weights.

**Update Historical States with Full Inference.** To enhance the tracking of intermediate hidden state attributes, we implement an exponential moving average during full inference on the selected weight channels $\mathbb{C}$. The update formula is expressed as:

$$\mathbf{V}_{:,\mathbb{C}}|^t = \lambda \mathbf{V}_{:,\mathbb{C}}|^{t-1} + (1-\lambda)||\tilde{\mathbf{X}}_{:,j,\mathbb{C}}^{\text{int}}||_2^2|^t, \tag{12}$$

The value of $\mathbf{V}$ is updated for $t$-th inference batch, and $\tilde{\mathbf{X}}^{\text{int}}$ represents the intermediate hidden states during full inference. We consistently set $\lambda = 0.99$ across all implementations.

## 4.4 PRUNING METRIC

We propose a new structured pruning metric named PPsp, where "sp" stands for structured pruning. This metric more effectively selects channels for pruning compared to existing metrics. We adapt the unstructured pruning metric Wanda (Sun et al., 2023) to a structured pruning scenario. PPsp introduces two enhancements: (1) we preserve the inherent importance of individual weights, as represented by the squared value of the Wanda metric; and (2) we calculate the $L^2$ norm of the importance scores for MLP input channels and attention heads to determine the pruning structures' importance, rather than summing these scores across pruning structures.

We introduce the pruning metric for a general scenario. To enhance clarity, we omit the superscript $l$, which denotes the block number, in this section. At each block, given intermediate hidden states $\mathbf{X}^{\text{int}}$ of shape $(N, S, C_{\text{in}})$, where $N$ and $S$ represent the batch and sequence dimensions respectively, and the weight matrix $\mathbf{W}^{\text{final}}$ of shape $(C_{\text{out}}, C_{\text{in}})$, Wanda (Sun et al., 2023) defines the importance of the individual weight $W_{i,k}^{\text{final}}$ as:

$$\mathbf{I}_{i,k} = |W_{i,k}^{\text{final}}| \cdot ||\mathbf{X}_{:,:,k}^{\text{int}}||_2, \tag{13}$$

where $|\cdot|$ denotes the absolute value operation, and $||\mathbf{X}^{\text{int}}_{:,:,k}||_2$ evaluates the $L^2$ norm of the $k$th feature across the $(N, S)$ dimensions. These two scalar values are then multiplied to produce the final importance. However, as derived in Wanda (Sun et al., 2023), the inherent importance of an individual weight is defined by:

$$\mathbf{I}_{i,k} = (|\boldsymbol{W}^{\text{final}}_{i,k}| \cdot ||\mathbf{X}^{\text{int}}_{:,:,k}||_2)^2 = |\boldsymbol{W}^{\text{final}}_{i,k}|^2 \cdot ||\mathbf{X}^{\text{int}}_{:,:,k}||_2^2. \tag{14}$$

Wanda discards the squaring in Eq. (14) in local weight importance ordering, as the non-negative nature of $|\boldsymbol{W}^{\text{final}}_{i,k}|$ and $||\mathbf{X}^{\text{int}}_{:,:,j}||_2$ does not impact the relative ordering of importance. However, when it comes to structured pruning, maintaining the inherent importance of individual weights is essential. Thus, we square the Wanda metric and compute the Euclidean distance across the $C_{\text{out}}$ dimension of the input channel. The formula is given by:

$$\mathbf{I}_k = \left\| \left\{ |\boldsymbol{W}^{\text{final}}_{i,k}|^2 \cdot ||\mathbf{X}^{\text{int}}_{:,:,k}||_2^2 \right\}_{i=0}^{C_{\text{out}}} \right\|_2, \tag{15}$$

where $\{\cdot\}$ signifies the set of elements, and $\mathbf{I} \in \mathbb{R}^{C_{\text{in}}}$.

# 5 EXPERIMENTAL SETUP

Table 1: Comparison of LLM structured pruning methods. Our implementation loads the full model for dynamic pruning, while other methods load only the pruned version.

| Method | No Fine-tuning | Time-Efficient | Easy Integration | Dynamic Pruning |
|---|---|---|---|---|
| Wanda-sp | ✓ | ✓ | ✓ | ✗ |
| FLAP | ✓ | ✓ | ✓ | ✗ |
| LLM-Pruner | ✗ | ✗ | ✗ | ✗ |
| LoRAPrune | ✗ | ✗ | ✗ | ✗ |
| PP | ✓ | ✓ | ✓ | ✓ |

We conduct three experiments using different random seeds for all tests and show the standard error across these three seeds in brackets. We conduct all experiments on NVIDIA A100 GPUs.

**Models and Evaluation.** We evaluate PP on three popular model families: LLaMA-2 7B/13B (Touvron et al., 2023), LLaMA-3 8B (Meta AI, 2024), and OPT-13B (Zhang et al., 2022). Following previous work (Sun et al., 2023; An et al., 2024), we evaluate the models on two zero-shot task categories. We evaluate accuracy on commonsense reasoning tasks, including BoolQ (Clark et al., 2019), PIQA (Bisk et al., 2020), HellaSwag (Zellers et al., 2019), WinoGrande (Sakaguchi et al., 2019), ARC-Easy (Clark et al., 2018), ARC-Challenge (Clark et al., 2018), and OpenbookQA (Mihaylov et al., 2018). For evaluating perplexity on the text generation task, we use WikiText2 (Merity et al., 2016). We set the batch size to 20 for all tasks. For the commonsense reasoning tasks, our implementation follows (Gao et al., 2021), setting the sequence length of each batch to match its longest sample. For the text generation task, we set the sequence length to 1024. For PP, we set the default probe size to 5% of the batch size and 50% of the sequence length, approximating 1.5% of the FLOPs cost relative to dense model inference. Figure 3 shows ablation study results for various probe combinations, indicating small probes enhance model performance. Ablation studies of the PP and FLAP are available in Appendix B, and additional experimental results are available in Appendix C.

**Baselines.** We compare our method, PP, with four previous approaches: Wanda-sp (An et al., 2024), FLAP (An et al., 2024), LoRAPrune (Zhang et al., 2023), and LLM-Pruner (Ma et al., 2023). We also compare PP with its upper bound, Full-Batch Probing, as introduced in Section 4.1. Following (Sun et al., 2023; An et al., 2024), we use the C4 (Raffel et al., 2020) dataset as the calibration dataset for all methods. We use 2,000 calibration samples for PP, Wanda-sp, and FLAP, and 20,000 calibration samples for tuning LoRAPrune and LLM-Pruner. We evaluate pruning ratios of 20% and 40%.

# 6 RESULTS

**Main Results.** We present the zero-shot performance, without fine-tuning, of four models on text generation and commonsense reasoning tasks, as shown in Tables 2 and 3. Probe Pruning (PP) consistently outperforms all baselines across various models and pruning ratios. For instance, on WikiText2 at a 40% pruning ratio, PP achieves lower perplexities than competing methods: 16.8 with LLaMA-2-7B, 11.3 with LLaMA-2-13B, and 26.7 with OPT-13B. Moreover, PP attains significantly lower perplexities and higher reasoning task accuracies than both LLM-Pruner and LoRAPrune. For example, on LLaMA-2-13B at a 40% pruning ratio, PP achieves an average accuracy of 61.0%, significantly higher than 52.0% for LLM-Pruner and 48.1% for LoRAPrune. On LLaMA-3-8B,

Table 2: Zero-shot performance of LLaMA-2-7B/13B and OPT-13B after pruning attention and MLP blocks without fine-tuning: PP demonstrates superior performance in nearly all scenarios. Arrows indicate metric direction (↓: lower is better; ↑: higher is better).

| Method | Pruning Ratio | Text Generation ↓ | | | Commonsense Reasoning ↑ | | |
|---|---|---|---|---|---|---|---|
| | | LLaMA-2-7B | LLaMA-2-13B | OPT-13B | LLaMA-2-7B | LLaMA-2-13B | OPT-13B |
| Dense | 0% | 6.0(0.1) | 5.1(0.1) | 11.6(0.1) | 64.0 | 66.2 | 57.2 |
| Full-Batch Probing | 20% | 7.3(0.1) | 6.2(0.1) | 12.6(0.1) | 62.6 | 65.3 | 56.4 |
| Wanda-sp | 20% | 10.6(0.1) | 9.0(0.1) | 17.4(0.1) | 61.5 | 65.0 | 55.2 |
| FLAP | 20% | 10.3(0.1) | 7.5(0.1) | 18.8(0.2) | 61.4 | 64.6 | 54.9 |
| LoRAPrune w/o LoRA | 20% | 22.7(0.9) | 16.1(0.7) | — | 57.9 | 58.9 | — |
| LLM-Pruner w/o LoRA | 20% | 17.5(1.6) | 11.3(0.7) | — | 57.4 | 61.3 | — |
| PP | 20% | **8.1(0.1)** | **6.7(0.1)** | **14.7(0.1)** | **62.8** | **65.3** | **56.5** |
| Full-Batch Probing | 40% | 13.6(0.1) | 8.9(0.1) | 17.9(0.2) | 58.7 | 62.9 | 54.0 |
| Wanda-sp | 40% | 43.8(1.5) | 21.6(0.4) | 42.7(0.7) | 54.8 | 56.6 | 50.5 |
| FLAP | 40% | 38.9(1.3) | 15.5(0.0) | 51.0(0.7) | 54.9 | 60.6 | 50.8 |
| LoRAPrune w/o LoRA | 40% | 129.5(3.0) | 74.8(6.4) | — | 45.4 | 48.1 | — |
| LLM-Pruner w/o LoRA | 40% | 51.1(4.3) | 34.5(2.4) | — | 47.8 | 52.0 | — |
| PP | 40% | **16.8(0.1)** | **11.3(0.1)** | **26.7(0.3)** | **56.6** | **61.0** | **53.1** |

Table 3: Zero-shot performance of LLaMA-3-8B after pruning MLP blocks without fine-tuning: PP demonstrates superior performance in nearly all scenarios.

| Method | Pruning Ratio | WikiText2 ↓ | BoolQ | PIQA | HellaSwag | WinoGrande | ARC-c | ARC-e | OBQA | Average ↑ |
|---|---|---|---|---|---|---|---|---|---|---|
| Dense | 0% | 6.8(0.0) | 81.7(0.0) | 79.5(0.0) | 76.3(0.0) | 72.5(0.0) | 47.2(0.0) | 61.7(0.0) | 40.2(0.0) | 65.6 |
| Full-Batch Probing | 20% | 8.5(0.0) | 79.0(0.0) | 80.1(0.0) | 74.8(0.0) | 73.9(0.0) | 44.9(0.0) | 60.7(0.0) | 40.2(0.0) | 64.8 |
| Wanda-sp | 20% | 10.0(0.0) | 75.1(0.3) | 78.5(0.0) | 69.6(0.2) | 71.4(0.4) | 38.7(0.4) | 56.9(0.4) | 39.0(0.2) | 61.3 |
| FLAP | 20% | 10.0(0.0) | **79.4(0.2)** | **78.7(0.1)** | 70.3(0.0) | 71.4(0.5) | 40.8(0.1) | 57.8(0.0) | 39.4(0.3) | 62.5 |
| PP | 20% | **9.3(0.0)** | 77.4(0.0) | 78.5(0.0) | **73.1(0.0)** | **72.5(0.3)** | **43.2(0.3)** | **59.1(0.2)** | **40.2(0.5)** | **63.4** |
| Full-Batch Probing | 40% | 12.3(0.1) | 73.1(0.0) | 77.8(0.0) | 70.5(0.0) | 70.3(0.0) | 42.9(0.0) | 58.9(0.0) | 39.8(0.0) | 61.9 |
| Wanda-sp | 40% | 18.4(0.1) | 66.6(0.1) | 73.4(0.2) | 56.7(0.1) | 63.2(0.2) | 31.8(0.2) | 47.0(0.5) | 34.5(0.2) | 53.3 |
| FLAP | 40% | 18.5(0.2) | 67.3(1.0) | 73.5(0.0) | 57.2(0.2) | 66.7(0.5) | 31.7(0.3) | 44.6(0.3) | 34.4(0.3) | 53.6 |
| PP | 40% | **14.9(0.1)** | **70.3(0.1)** | **76.3(0.2)** | **65.3(0.1)** | **67.2(0.2)** | **39.0(0.3)** | **57.4(0.1)** | **36.9(0.3)** | **58.9** |

PP surpasses Wanda-sp and FLAP in nearly all tasks, confirming its effectiveness and robustness. For instance, at a 40% pruning ratio, PP achieves an average accuracy of 58.9%, outperforming Wanda-sp (53.3%) and FLAP (53.6%). In Section 4.1, we stated that Full-Batch Probing represents the upper bound of PP. Experimental results confirm that Full-Batch Probing excels in all tested scenarios, supporting our hypothesis. Compared to Full-Batch Probing, which requires significant extra computational resources—more than dense model inference—PP achieves comparable results while utilizing minimal computational resources, only 1.5% of the FLOPs compared to dense model inference. These results imply the effectiveness of PP and demonstrate that the probe's intermediate hidden states can help identify the important weights for processing different batches.

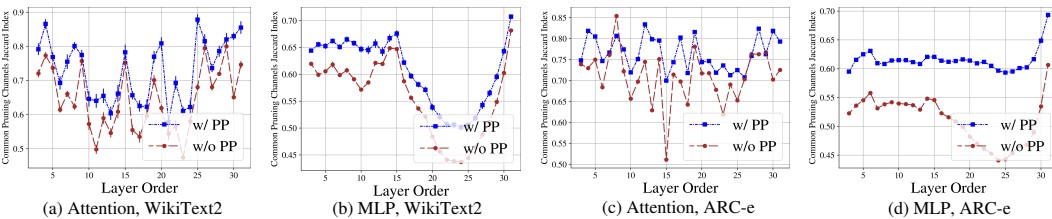

Figure 2: Jaccard Index of common pruning channels: comparing PP and Full-Batch Probing, and comparing fix-pruned model (without PP) and Full-Batch Probing for each batch.

**Jaccard Index of Common Pruning Channels.** To verify our assumption in Section 4.1 that a greater overlap of pruning channels between PP and Full-Batch Probing correlates with enhanced model performance and probe quality, we measure the Jaccard Index (Jaccard, 1912) of common pruning channels in two comparisons: between PP and Full-Batch Probing, and between the fix-pruned model (without PP) and Full-Batch Probing. The Jaccard Index is a statistical measure of the similarity between two sets, defined as the size of their intersection divided by the size of their union. We consistently apply the PPsp metric in all comparisons. As shown in Figure 2, PP consistently selects pruning channels more similar to those selected by Full-Batch Probing across almost all

attention and MLP blocks, in contrast to the fix-pruned model (without PP). This increased alignment of channels contributes to improved overall performance and indicates that the probe's intermediate hidden states can help guide pruning decisions.

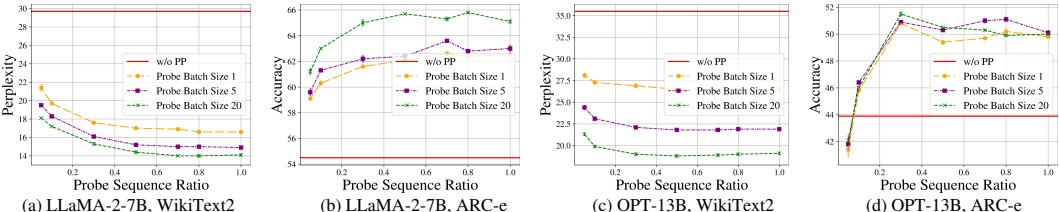

|  |  |  |  |
| --- | --- | --- | --- |
| (a) LLaMA-2-7B, WikiText2 | (b) LLaMA-2-7B, ARC-e | (c) OPT-13B, WikiText2 | (d) OPT-13B, ARC-e |

Figure 3: Performance of different probe combinations at a 40% pruning ratio.

**Effect of Probe Combinations on Performance.** We find that even a small probe can improve model performance. The results are shown in Figure 3. We investigate how different probe sizes affect PP's performance by varying the probe batch size from 1 to 20 (specifically, 1, 5, and 20) and the probe sequence ratio from 0.05 to 1.0 (specifically, 0.05, 0.1, 0.3, 0.5, 0.8, and 1.0). First, we observe that once we apply PP, even a small probe with a batch size of 1 and a probe sequence ratio of 0.05 can yield performance improvements. For example, for LLaMA-2-7B, the perplexity drops from 29.8 to 21.7; for OPT-13B, it drops from 35.4 to 27.7. Furthermore, we

Table 4: Comparison of FLOPs between dense model inference and probing.

| Method | Computational Cost (TFLOPs) | |
| --- | --- | --- |
|  | WikiText2 | ARC-c |
| Dense | 4420 | 4377 |
| Probing | 66 (1.5%) | 69 (1.6%) |

observe that increasing both the probe batch size and sequence ratio leads to improved performance. Interestingly, we find that the initial increase in sequence ratio from 0.05 to 0.3 brings the most rapid performance improvement. This indicates that sequence information becomes significantly effective for pruning once it exceeds a certain size threshold relative to the current batch's sequence length.

**Computational Cost and Inference Speed.** We use the DeepSpeed package (Rasley et al., 2020) to measure the FLOPs. The results in Table 4 show that the computational overhead of probing is approximating 1.5% of the FLOPs of the dense model inference. This finding aligns with our analyzed computational complexity in Section 4.1. Additionally, we evaluate each block's end-to-end runtime across all batches of WikiText2 and the inference speedup at a 40% pruning ratio on NVIDIA A100 GPUs, similar to previous

Table 5: Breakdown of inference runtime across all batches of WikiText2 at a 40% pruning ratio. The speedup is calculated by dividing the dense model's inference runtime by the methods' inference runtime.

| Method | PRR | Runtime (s) | | | |
| --- | --- | --- | --- | --- | --- |
|  |  | Attention | Speedup | MLP | Speedup |
| Dense | - | 0.612 | - | 0.416 | - |
| FLAP | 95.64 | 0.419 | 1.46× | 0.265 | 1.57× |
| Wanda-sp | 106.48 | 0.395 | 1.55× | 0.278 | 1.50× |
| PP | 37.37 | 0.420 | 1.46× | 0.319 | 1.30× |

studies (Sun et al., 2023; Ma et al., 2023). The results for LLaMA-2-7B on WikiText2 are presented in Table 5. We find that the inference speeds of PP are comparable to those of other structured pruning baselines, yet it delivers superior performance. Specifically, in the attention block, PP achieves a speedup of $1.46\times$, and in the MLP block, a speedup of $1.30\times$. The slight delay observed in the MLP block can be attributed to inherent system costs, such as weight extraction. This gap narrows under conditions with larger batch sizes or longer sequence lengths, leading to comparable speeds between PP and the baselines.

**Performance Runtime Ratio.** To illustrate the trade-off between model performance and inference speed, we introduce Performance Runtime Ratio (PRR), which quantifies the ratio of performance degradation per unit of runtime reduction. Importantly, a *smaller* PRR value is preferable as it indicates minimal performance degradation per unit of runtime reduction. The metric is defined as:

$$\text{PRR} = \frac{|\text{Perf}_{\text{dense}} - \text{Perf}_{\text{pruned}}|}{\text{Runtime}_{\text{dense}} - \text{Runtime}_{\text{pruned}}}, \tag{16}$$

where $\text{Perf}_{\text{pruned}}$ and $\text{Runtime}_{\text{pruned}}$ denote the performance and runtime of the pruned model, respectively, and $\text{Perf}_{\text{dense}}$ and $\text{Runtime}_{\text{dense}}$ denote the performance and runtime of the dense model, respectively. As shown in Table 5, the PRR of PP is 37.37, indicating a increase of 37.37 in perplexity per second of runtime reduction on the attention and MLP block. In comparison, FLAP and Wanda-sp

have PRR values of 95.65 and 106.48, respectively. PP's PRR values are 2.56× (95.65 compared to 37.37) and 2.85× (106.48 compared to 37.37) more efficient than those of FLAP and Wanda-sp, respectively, indicating a significantly lower rate of performance degradation.

**Compared with Fine-tuned Baselines.** Table 6 compares the performance of PP with fine-tuned baselines LoRAPrune and LLM-Pruner across different pruning ratios for text generation and commonsense reasoning tasks. Without fine-tuning, PP consistently outperforms or closely matches the fine-tuned models. At a 20% pruning ratio, PP excels in both tasks across LLaMA-2-7B and LLaMA-2-13B models. At a 40% pruning ratio, PP achieves comparable perplexity and significantly higher reasoning task accuracies. For example, PP achieves 61 on LLaMA-2-13B, while LoRAPrune achieves 55.5 and LLM-Pruner achieves 54.7.

Table 6: Comparison of PP with fine-tuned baselines on LLaMA-2-7B/13B models, with attention and MLP layers pruned: PP consistently outperforms across scenarios without fine-tuning.

| Method | Pruning Ratio | Fine-tuning | Text Generation ↓ | | Commonsense Reasoning ↑ | |
| --- | --- | --- | --- | --- | --- | --- |
| | | | LLaMA-2-7B | LLaMA-2-13B | LLaMA-2-7B | LLaMA-2-13B |
| Dense | 0% | ✗ | 6.0(0.1) | 5.1(0.1) | 64.0 | 66.2 |
| LoRAPrune w/ LoRA | 20% | ✓ | 8.7(0.2) | 7.4(0.0) | 59.2 | 61.0 |
| LLM-Pruner w/ LoRA | 20% | ✓ | 10.2(0.3) | 8.4(0.5) | 58.7 | 62.1 |
| PP | 20% | ✗ | **8.1(0.1)** | **6.7(0.1)** | **62.8** | **65.3** |
| LoRAPrune w/ LoRA | 40% | ✓ | **13.6(0.4)** | **11.1(0.3)** | 52.1 | 55.5 |
| LLM-Pruner w/ LoRA | 40% | ✓ | 20.3(1.3) | 15.3(0.7) | 50.6 | 54.7 |
| PP | 40% | ✗ | 16.8(0.1) | 11.3(0.1) | **56.6** | **61.0** |

**Importance-Scaled Fusion.** We compare importance-scaled fusion to three fixed integration ratios—0.1, 0.5, and 0.9—which assign a fixed ratio to the probing states during integration with historical states. We conduct experiments on LLaMA-2-7B using the WikiText2 dataset at a 40% pruning ratio, keeping the probe batch size fixed at 1. The results in Figure 4 demonstrate that importance-scaled fusion can leverage the benefits of the calibration dataset while minimizing associated biases.

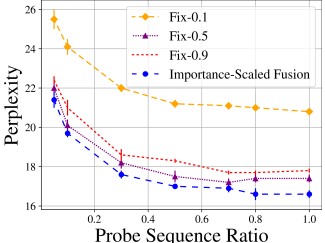

Figure 4: Importance-scaled fusion studies.

**Pruning Metric.** Our PPsp consistently outperforms both Wanda-sp and FLAP across various pruning scenarios. We conduct experiments on fix-pruned models, each uniquely generated by one of three evaluated metrics, using only the calibration dataset. we evaluated three metrics at a uniform 40% pruning ratio across all blocks on the WikiText2 dataset. As shown in Table 7, PPsp significantly reduces perplexity, achieving the lowest scores of 29.7 and 35.5 on the LLaMA-2-7B and OPT-13B models, respectively, compared to FLAP's 38.2 and 41.1, and Wanda-sp's 43.8 and 42.7.

Table 7: Perplexity of WikiText2 across different metrics on models pruned by the calibration dataset, showing that PPsp performs best among the three metrics.

| Metric | Formula | LLaMA-2-7B | | | OPT-13B | | |
| --- | --- | --- | --- | --- | --- | --- | --- |
| | | Attention | MLP | All | Attention | MLP | All |
| Wanda-sp | $\sum_{i=1}^{C_{out}} |W_{i,k}^{final}| \cdot ||\mathbf{X}_{:,:,k}^{int}||_2$ | 21.1(0.2) | **10.9(0.1)** | 43.8(1.5) | 13.2(1.3) | 27.5(0.4) | 42.7(0.7) |
| FLAP | $\frac{1}{N-1}\sum_{n=1}^{N} ||W_{:,k}^{final}||_2^2 \cdot (\mathbf{X}_{n,:,k}^{int} - \overline{\mathbf{X}_{:,:,k}^{int}})^2$ | 17.7(0.3) | 11.0(0.1) | 38.2(0.3) | **11.6(0.1)** | 27.3(0.0) | 41.1(0.3) |
| PPsp | $\left\|\left\{ |W_{i,k}^{final}|^2 \cdot ||\mathbf{X}_{:,:,k}^{int}||_2^2 \right\}_{i=0}^{C_{out}}\right\|_2$ | **15.4(0.6)** | **10.9(0.1)** | **29.7(0.3)** | 12.9(1.0) | **25.1(0.3)** | **35.5(0.3)** |

## 7 CONCLUSION

In this paper, we propose Probe Pruning (PP), a novel online dynamic pruning framework that uses a *small yet crucial* portion of hidden states to run the model and gain crucial pruning information to guide full inference. Notably, PP only relies on the original model structure and hidden states, *without requiring additional neural network modules, or fine-tuning.* Furthermore, PP consistently surpasses all baselines, including those with fine-tuning in almost all experimental settings. Future research directions include refining the probe generation and probing process, integrating PP with advanced decoding and alignment techniques (Wang et al., 2025), and exploring its robustness against poisoned models (Xian et al., 2023a;b; Wang et al., 2024a) or adversarial prompts (Xian et al., 2025).

ACKNOWLEDGMENT

The authors acknowledge the Minnesota Supercomputing Institute (MSI) at the University of Minnesota for providing resources that contributed to the research results reported in this paper. URL: http://www.msi.umn.edu.

The work of Qi Le was supported by the Amazon Machine Learning System Fellowship. The work of Xinran Wang and Ali Anwar was supported by the 3M Science and Technology Graduate Fellowship and the Samsung Global Research Outreach Award. The work of Jie Ding was supported in part by the National Science Foundation under CAREER Grant No. 2338506 and Grant No. 2220286. The work of Ziyan Wang and Li Yang was supported by National Science Foundation under Grant No. 2348376.

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

# Appendix for "Probe Pruning"

## A  IMPLEMENTATION DETAILS

For all methods, we leave the first three layers unchanged, similar to Ma et al. (2023); Zhang et al. (2023), because pruning parameters in these layers has a substantial impact on the model. The pruning ratio represents the average pruning ratio across all attention and MLP blocks in the model. For instance, when targeting pruning ratios of 20% and 40% for LLaMA-2-7B, we prune 22% and 44% from attention and MLP blocks 4 to 32, respectively.

For a fair comparison, we utilize the exact same subset of the C4 Raffel et al. (2020) dataset as the calibration dataset.

For PP, FLAP An et al. (2024), and Wanda-sp An et al. (2024), we use 2,000 samples with sequence lengths of 1,024 tokens as the calibration dataset for the text generation task, and 2,000 samples with sequence lengths of 512 tokens for the commonsense reasoning task.

For LLM-Pruner Ma et al. (2023), we follow the original implementation details in Ma et al. (2023). We use 10 randomly selected samples, each truncated to a length of 128 tokens, to build importance metrics, and 20,000 samples with sequence lengths of 256 tokens for recovery retraining. Specifically, in the recovery stage, we employ the AdamW He et al. (2020) optimizer with 100 warmup steps, set the LoRA Hu et al. (2021) rank $r$ to 8, use a learning rate of $1 \times 10^{-4}$, a batch size of 64, and perform recovery retraining for 2 epochs.

For LoRAPrune Zhang et al. (2023), we follow the original implementation details in Zhang et al. (2023). We randomly sample 20,000 sentences from the C4 dataset, each having a length of 512 tokens, according to the original calibration dataset preparation process. The training hyperparameters include setting the LoRA rank to 8, a learning rate of $1 \times 10^{-4}$, a batch size of 128, and a total of 2 training epochs. When fusing pruning with fine-tuning, we employ a cubic sparsity scheduler Sanh et al. (2020) to iteratively prune the model until we reach the target sparsity. When only pruning is performed, with no tuning conducted to match other one-shot pruning methods, we use 10 selected samples with sequence lengths of 512 tokens to estimate importance and perform one-shot pruning with no weight updates. All training processes are optimized using the AdamW optimizer with a linear learning rate decay.

## B  ABLATION STUDIES

In this section, we present various ablation studies. Section B.1 investigates how different calibration datasets influence the fix-pruned model, which relies exclusively on such calibration dataset. Section B.2 evaluates the effect of manually squaring the attention metric in the FLAP model An et al. (2024) versus not squaring it. Section B.3 illustrates the batch-dependent outliers at token positions. Section B.4 studies the effectiveness of residual importance. Section B.5 studies the integration of historical states and their influence on the performance of Probe Pruning (PP). Section B.6 verifies the possibility of running the probe in parallel with the actual computation of earlier pruned blocks. Section B.7 analyzes the discrepancies between pruning the attention and MLP blocks at varying pruning ratios.

### B.1  CALIBRATION DATASET

We present the performance of FLAP An et al. (2024) using different calibration datasets to test WikiText2 Perplexity, as shown in Table 8. The results indicate that structured pruning methods, which rely solely on calibration datasets, may introduce biases. For instance, when using the WikiText2 validation set as a calibration dataset, FLAP achieves a perplexity of 18.5 at a 40% pruning ratio on WikiText2. However, with the C4 dataset as the calibration dataset, the perplexity deteriorates to 38.9.

### B.2  MANUALLY SQUARING THE ATTENTION METRIC

In the FLAP implementation available at `https://github.com/CASIA-IVA-Lab/FLAP`, the attention metric is manually squared. Table 9 demonstrates the impact of manually squaring the

Table 8: Comparison of FLAP performance at different pruning ratios and calibration datasets on LLaMA-2-7B and LLaMA-2-13B models.

| Method | Pruning Ratio | Calibration Dataset | LLaMA-2-7B | LLaMA-2-13B |
|--------|---------------|---------------------|------------|-------------|
| FLAP | 20% | C4 | 10.30(0.1) | 7.5(0.1) |
| | 20% | WikiText2 - validation | **7.9(0.1)** | **6.5(0.1)** |
| | 40% | C4 | 38.9(1.3) | 15.5(0.0) |
| | 40% | WikiText2 - validation | **18.5(0.2)** | **10.5(0.1)** |

Table 9: Comparasion of FLAP with and without squaring the attention metric, while keeping the MLP metric consistently unsquared, on LLaMA-2-7B and LLaMA-2-13B Models.

| | | LLaMA-2-7B | | | LLaMA-2-13B | | |
|--------|---------------|-------------------------|--------------------|----------|-------------------------|--------------------|----------|
| Method | Pruning Ratio | Attention Pruning Ratio | MLP Pruning Ratio | WikiText2 | Attention Pruning Ratio | MLP Pruning Ratio | WikiText2 |
| FLAP w/o square | 20% | 17.8%(0.1) | 21.3%(0.1) | 10.3(0.1) | 24.7%(0.1) | 18.0%(0.1) | **7.5(0.1)** |
| FLAP | 20% | 0.6%(0.1) | 30.8%(0.1) | **9.1(0.1)** | 0.0%(0.0) | 31.5%(0.1) | 7.7(0.1) |
| FLAP w/o square | 40% | 35.4%(0.1) | 42.6%(0.1) | 38.9(1.3) | 37.5%(0.1) | 41,0%(0.1) | 15.5(0.0) |
| FLAP | 40% | 17.6%(0.1) | 52.6%(0.1) | **29.1(0.4)** | 11.4%(0.1) | 55.6%(0.1) | **13.6(0.1)** |

attention metric in FLAP versus not squaring it. The findings indicate that squaring the metric results in less aggressive pruning of attention blocks. For instance, with LLaMA-2-7B at a 20% overall pruning ratio, the non-squared FLAP method prunes 17.8% of attention weights, in contrast to only 0.6% when squaring is implemented. This implies that squaring significantly mitigates attention pruning.

Additionally, less aggressive pruning of attention blocks correlates with better model performance. Specifically, on LLaMA-2-7B at a 40% overall pruning ratio, non-squared FLAP prunes 35.4% of attention weights, resulting in a WikiText2 perplexity of 38.9. Conversely, squared FLAP prunes at a reduced rate of 17.6%, achieving a lower perplexity of 29.1. These outcomes suggest that more conservative pruning of attention blocks can enhance model performance.

## B.3 BATCH-DEPENDENT OUTLIERS AT TOKEN POSITIONS

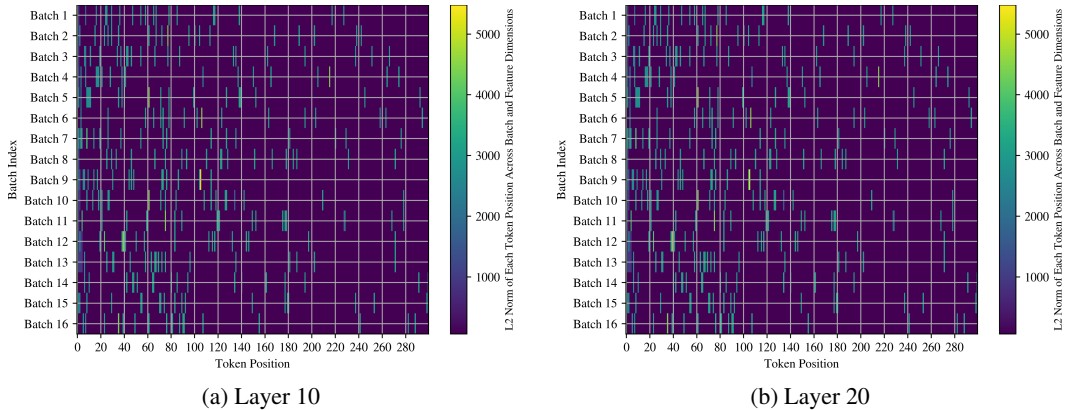

(a) Layer 10        (b) Layer 20

Figure 5: Visualization of the $L^2$ norm for each token position of the input hidden states at layers 10 and 20 across the batch and feature dimensions. Experiments are conducted on the LLaMA-2-7B model using the WikiText2 dataset.

In the main text Section 1, we stated that large language models (LLMs) exhibit batch-dependent outliers, necessitating online dynamic pruning to address these dynamic outliers. Figure 5 presents the calculated $L^2$ norms for each token position of the input hidden states at layers 10 and 20 across the batch and feature dimensions. The results demonstrate the presence of batch-dependent outliers

at each token position, aligning with the observations from existing works Liu et al. (2024); Sun et al. (2024).

## B.4 RESIDUAL IMPORTANCE

In the main text Section 4.2, we noted that layer normalization significantly alters the input hidden states, thereby preventing their importance from accurately identifying key samples and tokens. To validate this observation, Table 10 compares the effectiveness of identifying key samples and tokens based on residual importance with identification based on the importance of layer-normalized input hidden states (PP without residual importance). The experimental results demonstrate the effectiveness of residual importance.

Table 10: Impact of residual importance on probe generation for LLaMA-2-7B. Applying residual importance results in better probe performance.

| Method | Pruning Ratio | WikiText2 | BoolQ | PIQA | HellaSwag | WinoGrande | ARC-c | ARC-e | OBQA | Average |
|---|---|---|---|---|---|---|---|---|---|---|
| PP w/o residual importance | 20% | 10.3(0.0) | 64.3(0.1) | 74.2(0.2) | 55.3(0.1) | 53.4(0.5) | 32.1(0.2) | 55.6(0.1) | 40.2(0.2) | 53.6 |
| PP | 20% | **8.1(0.1)** | **69.0(0.1)** | **78.1(0.0)** | **73.5(0.0)** | **66.7(0.3)** | **42.8(0.1)** | **68.5(0.0)** | **40.9(0.2)** | **62.8** |
| PP w/o residual importance | 40% | 37.1(0.4) | 62.1(0.0) | 61.1(0.0) | 31.0(0.0) | 50.2(0.1) | 20.4(0.2) | 34.4(0.2) | 36.7(0.3) | 42.3 |
| PP | 40% | **16.8(0.1)** | **62.7(0.2)** | **74.9(0.1)** | **63.6(0.0)** | **57.5(0.2)** | **35.5(0.1)** | **61.7(0.2)** | **40.3(0.4)** | **56.6** |

## B.5 HISTORICAL STATES INTEGRATION

In Table 11, the results illustrate how incorporating historical states into the pruning decision process enhances the effectiveness of PP. Specifically, when PP leverages historical states, there is a consistent improvement in performance metrics across all models and pruning ratios compared to scenarios where only probing states are utilized (PP w/o historical states). For instance, at a 40% pruning ratio, using a probe generated from 5% of the batch and 50% of the sequence, PP with historical states reduces the perplexity on WikiText2 from 20.1 to 16.9 and improves the average accuracy from 51.2% to 56.6%, compared to using only the current probing states without historical data.

Table 11: Performance of integrating historical states under different probe combinations on LLaMA-2-7B. historical states can enhance PP performance.

| Method | Probe Generation | Pruning Ratio | WikiText2 | BoolQ | PIQA | HellaSwag | WinoGrande | ARC-c | ARC-e | OBQA | Average |
|---|---|---|---|---|---|---|---|---|---|---|---|
| PP w/o historical states | 5% batch, 50% seq | 20% | 8.2(0.1) | 68.4(0.0) | 75.8(0.0) | 70.4(0.0) | 63.2(0.0) | 38.9(0.0) | 64.4(0.0) | 42.2(0.0) | 60.5 |
| PP w/o historical states | 10% batch | 20% | 7.9(0.1) | 69.8(0.0) | 75.7(0.0) | 70.7(0.0) | 63.7(0.0) | 39.2(0.0) | 64.9(0.0) | 41.4(0.0) | 60.8 |
| PP w/o historical states | 20% batch | 20% | 7.7(0.1) | 69.3(0.0) | 76.7(0.0) | 70.9(0.0) | 63.8(0.0) | 40.1(0.0) | 65.4(0.0) | 40.2(0.0) | 60.9 |
| PP | 5% batch, 50% seq | 20% | 8.2(0.1) | 69.0(0.1) | 78.1(0.0) | 73.5(0.0) | 66.7(0.3) | 42.8(0.1) | 68.5(0.0) | 40.9(0.2) | 62.8 |
| PP | 10% batch | 20% | 8.0(0.1) | 67.3(0.1) | 77.8(0.1) | 73.7(0.0) | 64.8(0.1) | 41.5(0.1) | 67.4(0.2) | 41.3(0.3) | 62 |
| PP | 20% batch | 20% | 7.8(0.1) | 68.1(0.1) | 77.5(0.1) | 73.7(0.0) | 66.7(0.3) | 42.2(0.1) | 68.2(0.1) | 42.7(0.4) | 62.7 |
| PP w/o historical states | 5% batch, 50% seq | 40% | 20.1(0.3) | 57.4(0.0) | 71.3(0.0) | 55.7(0.0) | 54.6(0.0) | 31.7(0.0) | 53.3(0.0) | 34.6(0.0) | 51.2 |
| PP w/o historical states | 10% batch | 40% | 17.2(0.4) | 62.1(0.0) | 72.1(0.0) | 56.9(0.0) | 58.3(0.0) | 34.3(0.0) | 57.9(0.0) | 35.4(0.0) | 53.9 |
| PP w/o historical states | 20% batch | 40% | 15.6(0.2) | 63.8(0.0) | 72.3(0.0) | 57.6(0.0) | 56.5(0.0) | 33.5(0.0) | 57.7(0.0) | 36.0(0.0) | 53.9 |
| PP | 5% batch, 50% seq | 40% | 16.9(0.1) | 62.7(0.2) | 74.9(0.1) | 63.6(0.0) | 57.5(0.2) | 35.5(0.1) | 61.7(0.2) | 40.3(0.4) | 56.6 |
| PP | 10% batch | 40% | 15.8(0.3) | 64.3(0.1) | 74.5(0.1) | 64.2(0.1) | 57.9(0.4) | 37.6(0.1) | 62.9(0.2) | 40.7(1.1) | 57.4 |
| PP | 20% batch | 40% | 15.1(0.2) | 64.7(0.1) | 74.3(0.1) | 64.4(0.1) | 58.1(0.3) | 37.7(0.3) | 62.5(0.1) | 41.3(0.2) | 57.6 |

## B.6 PARALLEL PROBING

We verify the possibility of running the probe in parallel with the actual computation of earlier pruned blocks. We present the results in Table 12 below. Here, **PP-Parallel** represents the approach where, when the actual computation happens on earlier pruned blocks, we generate the probe from the residuals of these earlier pruned blocks and perform the probing. **PP** represents the default setting used in the main text of the paper. The results show that we can still obtain performance gains and achieve comparable results to PP. For example, at a 40% pruning ratio, PP-Parallel achieves a perplexity of 17.9 on WikiText2, which is close to that of PP and much lower than the 38.9 achieved by FLAP. Furthermore, PP-Parallel achieves 61.4% accuracy on ARC-e, which is close to that of PP and much higher than the 52.5% achieved by FLAP. However, we are just demonstrating the feasibility of further improving PP's inference speed here; the actual parallelism is hardware-dependent and implementation-dependent.

Table 12: Zero-shot performance of LLaMA-2-7B after pruning attention and MLP blocks without fine-tuning.

| Method | Pruning Ratio | WikiText2 | ARC-e |
|---|---|---|---|
| Dense | 0% | 6.0(0.1) | 67.3(0.0) |
| Full-Batch Probing | 20% | 7.3(0.1) | 67.2(0.0) |
| Wanda-sp | 20% | 10.6(0.1) | 63.9(0.3) |
| FLAP | 20% | 10.3(0.1) | 63.1(0.1) |
| PP-Parallel | 20% | **8.1(0.1)** | 67.9(0.1) |
| PP | 20% | **8.1(0.1)** | **68.5(0.0)** |
| Full-Batch Probing | 40% | 13.6(0.1) | 64.7(0.0) |
| Wanda-sp | 40% | 43.8(1.5) | 54.4(0.1) |
| FLAP | 40% | 38.9(1.3) | 52.5(0.2) |
| PP-Parallel | 20% | 17.9(0.1) | 61.4(0.2) |
| PP | 40% | **16.8(0.1)** | **61.7(0.2)** |

Table 13: Performance of pruning attention heads versus MLPs at different ratios on LLaMA-2-7B, comparing the effects of pruning only the attention heads or only the MLPs.

| Pruning Ratio | | | Text Generation ↓ | Commonsense Reasoning ↑ | | | | | | | |
|---|---|---|---|---|---|---|---|---|---|---|---|
| Attention | MLP | All | WikiText2 | BoolQ | PIQA | HellaSwag | WinoGrande | ARC-c | ARC-e | OBQA | Average |
| 0% | 0% | 0% | 6.0(0.1) | 74.6(0.0) | 77.9(0.0) | 75(0.0) | 67.7(0.0) | 42.7(0.0) | 67.3(0.0) | 42.6(0.0) | 64.0 |
| 20% | 0% | 7% | **6.8(0.1)** | **71.1(0.1)** | **78.6(0.1)** | **74.7(0.0)** | 66.3(0.1) | **42.9(0.0)** | **69.0(0.1)** | **43.1(0.1)** | 63.7 |
| 0% | 20% | **13%** | 7.2(0.1) | 68.4(0.1) | 77.7(0.0) | 74.3(0.0) | **67.8(0.1)** | 41.8(0.2) | 66.9(0.1) | 41.3(0.1) | 62.6 |
| 40% | 0% | 14% | **10.0(0.0)** | 65.3(0.1) | **77.2(0.1)** | 69.3(0.1) | 58.4(0.2) | **38.1(0.1)** | **64.9(0.0)** | 40.3(0.1) | 59.1 |
| 0% | 40% | **25%** | 10.1(0.1) | **65.9(0.0)** | 76.0(0.2) | **69.4(0.1)** | **63.8(0.0)** | 36.5(0.2) | 62.4(0.0) | 40.3(0.6) | **59.2** |
| 60% | 0% | 21% | 33.5(0.4) | 60.8(0.1) | **71.4(0.1)** | 42.2(0.1) | 51.8(0.3) | 29.9(0.2) | 49.8(0.1) | **36.4(0.2)** | 49.0 |
| 0% | 60% | **39%** | **21.1(0.2)** | **62.8(0.1)** | 71.1(0.2) | **55.3(0.1)** | **58.6(0.3)** | **31.4(0.2)** | **53.4(0.0)** | 34.8(0.2) | **52.5** |
| 40% | 20% | 27% | 11.9(0.1) | 65.0(0.1) | **76.4(0.1)** | 68.4(0.1) | 59.3(0.4) | **39.0(0.1)** | **64.8(0.2)** | 40.6(0.3) | 59.1 |
| 20% | 40% | **33%** | **11.5(0.1)** | **67.7(0.1)** | 75.4(0.3) | **69.1(0.0)** | **62.7(0.2)** | 38.3(0.1) | 64.1(0.2) | **41.0(0.4)** | **59.8** |
| 60% | 20% | 34% | 38.4(0.3) | 62.0(0.1) | **72.6(0.2)** | 43.7(0.1) | 51.0(0.2) | 29.9(0.1) | 52.3(0.2) | **38.5(0.4)** | 50.0 |
| 20% | 60% | **46%** | **23.8(0.4)** | **62.5(0.1)** | 70.8(0.2) | **55.6(0.1)** | **58.5(0.2)** | **33.2(0.2)** | **54.6(0.1)** | 36.2(0.1) | **53.1** |
| 60% | 40% | 47% | 44.3(0.5) | **62.0(0.0)** | 70.8(0.2) | 42.8(0.1) | 51.0(0.4) | 29.0(0.2) | 51.2(0.2) | **36.9(0.5)** | 49.1 |
| 40% | 60% | **53%** | **33.5(1.2)** | 60.6(0.1) | 70.3(0.1) | **50.7(0.0)** | **53.8(0.3)** | **30.1(0.5)** | **52.8(0.2)** | 35.9(0.1) | **50.6** |

## B.7 DISCREPENCY BETWEEN PRUNING ATTENTION AND MLP.

We find that the pruning ratios for attention and MLP layers should be considered independently, as they may reach saturation at different points. Table 13 demonstrates a clear discrepancy in performance between pruning attention heads and MLPs, especially as the pruning ratios increase. While lower pruning ratios (20%) result in similar performance impacts for both components, higher ratios (40%, 60%) suggest that attention heads reach saturation, particularly in demanding tasks such as WikiText2 and HellaSwag. For example, at a 60% pruning ratio for attention, performance on WikiText2 drops dramatically to 33.5, compared to 21.1 when the MLP is pruned at the same level. Similarly, performance on HellaSwag decreases significantly to 42.2 when pruning attention, compared to 55.3 when pruning the MLP at the same level. Additionally, considering each module's actual FLOPs reveals a larger performance gap, emphasizing the need for a strategic approach to pruning neural network components.

## C ADDITIONAL EXPERIMENTAL RESULTS

In this section, we present the detailed experimental results for each task. The performance without fine-tuning is shown in Tables 14, 15, 16, and 17. The comparison of PP with fine-tuned baselines is provided in Tables 18 and 19. PP consistently surpasses all baselines, including those with fine-tuning, in almost all experimental settings.

Table 14: Zero-shot performance of LLaMA-2-7B after pruning attention and MLP blocks without fine-tuning: PP demonstrates superior performance in nearly all scenarios.

| Method | Pruning Ratio | WikiText2 | BoolQ | PIQA | HellaSwag | WinoGrande | ARC-c | ARC-e | OBQA | Average |
|---|---|---|---|---|---|---|---|---|---|---|
| Dense | 0% | 6.0(0.1) | 74.6(0.0) | 77.9(0.0) | 75(0.0) | 67.7(0.0) | 42.7(0.0) | 67.3(0.0) | 42.6(0.0) | 64.0 |
| Full-Batch | 20% | 7.3(0.1) | 67.9(0.0) | 77.0(0.0) | 74.5(0.0) | 65.9(0.0) | 42.7(0.0) | 67.2(0.0) | 43.2(0.0) | 62.6 |
| Wanda-sp | 20% | 10.6(0.1) | 65.3(0.1) | 77.2(0.1) | **74.1(0.0)** | 67.1(0.2) | 41.1(0.1) | 63.9(0.3) | 41.8(0.2) | 61.5 |
| FLAP | 20% | 10.3(0.1) | 67.3(0.5) | 76.6(0.2) | 73.0(0.1) | **67.4(0.0)** | 40.6(0.3) | 63.1(0.1) | **42.0(0.1)** | 61.4 |
| LoRAPrune | 20% | 22.7(0.9) | 64.2(0.6) | 74.6(0.3) | 66.5(0.5) | 58.8(1.2) | 37.7(0.7) | 63.9(0.6) | 39.4(1.1) | 57.9 |
| LLM-Pruner | 20% | 17.5(1.6) | 62.5(0.3) | 75.3(0.8) | 66.0(0.7) | 57.2(1.7) | 37.7(1.0) | 62.4(0.7) | 40.5(0.2) | 57.4 |
| PP | 20% | **8.1(0.1)** | **69.0(0.1)** | **78.1(0.0)** | 73.5(0.0) | 66.7(0.3) | **42.8(0.1)** | **68.5(0.0)** | 40.9(0.2) | **62.8** |
| Full-Batch | 40% | 13.6(0.1) | 64.8(0.0) | 74.9(0.0) | 67.6(0.0) | 59.0(0.0) | 38.7(0.0) | 64.7(0.0) | 41.0(0.0) | 58.7 |
| Wanda-sp | 40% | 43.8(1.5) | 62.5(0.1) | 72.5(0.1) | 63.3(0.0) | 56.9(0.1) | 33.4(0.2) | 54.4(0.1) | **40.8(0.4)** | 54.8 |
| FLAP | 40% | 38.9(1.3) | **63.5(0.1)** | 71.7(0.3) | 63.3(0.1) | **59.8(0.1)** | 33.8(0.6) | 52.5(0.2) | 40.0(0.6) | 54.9 |
| LoRAPrune | 40% | 129.5(3.0) | 54.0(4.2) | 65.0(0.5) | 45.1(1.3) | 52.1(0.3) | 25.8(0.2) | 43.6(0.7) | 32.1(0.6) | 45.4 |
| LLM-Pruner | 40% | 51.1(4.3) | 55.5(5.0) | 69.8(1.1) | 49.6(2.1) | 51.2(0.3) | 27.8(0.6) | 46.0(2.0) | 35.0(0.5) | 47.8 |
| PP | 40% | **16.8(0.1)** | 62.7(0.2) | 74.9(0.1) | **63.6(0.0)** | 57.5(0.2) | **35.5(0.1)** | **61.7(0.2)** | 40.3(0.4) | **56.6** |

Table 15: Zero-shot performance of LLaMA-2-13B after pruning attention and MLP blocks without fine-tuning: PP demonstrates superior performance in nearly all scenarios.

| Method | Pruning Ratio | WikiText2 | BoolQ | PIQA | HellaSwag | WinoGrande | ARC-c | ARC-e | OBQA | Average |
|---|---|---|---|---|---|---|---|---|---|---|
| Dense | 0% | 5.1(0.1) | 72.1(0.0) | 79.6(0.0) | 78.7(0.0) | 70.7(0.0) | 46.5(0.0) | 71.3(0.0) | 44.2(0.0) | 66.2 |
| Full-Batch | 20% | 6.2(0.1) | 69.0(0.0) | 78.7(0.0) | 77.9(0.0) | 70.1(0.0) | 47.7(0.0) | 71.1(0.0) | 42.8(0.0) | 65.3 |
| Wanda-sp | 20% | 9.0(0.1) | 70.4(1.0) | 79.4(0.1) | **78.4(0.0)** | 70.2(0.1) | 44.3(0.6) | 69.9(0.3) | 42.5(0.2) | 65.0 |
| FLAP | 20% | 7.5(0.1) | 71.1(0.5) | 78.7(0.1) | 77.3(0.0) | **71.2(0.2)** | 44.6(0.1) | 66.7(0.1) | 42.5(0.2) | 64.6 |
| LoRAPrune | 20% | 16.1(0.7) | 63.6(0.2) | 75.4(0.1) | 69.4(0.8) | 63.6(0.3) | 37.6(0.4) | 62.6(0.7) | 40.3(0.5) | 58.9 |
| LLM-Pruner | 20% | 11.3(0.7) | 63.4(1.8) | 77.7(0.1) | 72.3(0.5) | 63.0(1.1) | 42.3(0.6) | 67.8(0.3) | 42.9(0.7) | 61.3 |
| PP | 20% | **6.7(0.1)** | 72.0(0.2) | 79.5(0.1) | 77.6(0.0) | 68.5(0.1) | **44.7(0.2)** | 71.5(0.1) | 43.0(0.2) | **65.3** |
| Full-Batch | 40% | 8.9(0.1) | 68.4(0.0) | 77.7(0.0) | 74.5(0.0) | 65.4(0.0) | 42.4(0.0) | 69.3(0.0) | 42.8(0.0) | 62.9 |
| Wanda-sp | 40% | 21.6(0.4) | 62.4(0.0) | 74.5(0.3) | 68.0(0.0) | 63.0(0.4) | 34.8(0.5) | 54.9(0.3) | 38.9(0.4) | 56.6 |
| FLAP | 40% | 15.5(0.0) | 62.9(0.1) | 76.8(0.3) | **72.4(0.1)** | **66.9(0.3)** | 40.4(0.4) | 63.1(0.4) | 41.8(0.1) | 60.6 |
| LoRAPrune | 40% | 74.8(6.4) | 57.9(3.5) | 66.8(0.9) | 51.5(0.6) | 53.6(0.5) | 28.5(0.3) | 46.0(0.8) | 32.4(1.2) | 48.1 |
| LLM-Pruner | 40% | 34.5(2.4) | 57.0(2.2) | 72.5(1.1) | 57.8(2.0) | 54.2(0.8) | 33.3(1.3) | 51.5(1.9) | 37.7(1.2) | 52.0 |
| PP | 40% | **11.3(0.1)** | 65.8(0.1) | 77.1(0.2) | 71.6(0.0) | 61.3(0.4) | **40.9(0.3)** | 67.9(0.1) | 42.5(0.3) | **61.0** |

Table 16: Zero-shot performance of OPT-13B after pruning attention and MLP blocks without fine-tuning: PP demonstrates superior performance in nearly all scenarios.

| Method | Pruning Ratio | WikiText2 | BoolQ | PIQA | HellaSwag | WinoGrande | ARC-c | ARC-e | OBQA | Average |
|---|---|---|---|---|---|---|---|---|---|---|
| Dense | 0% | 11.6(0.1) | 68.1(0.0) | 75.3(0.0) | 67.9(0.0) | 66.8(0.0) | 35(0.0) | 51.1(0.0) | 36.4(0.0) | 57.2 |
| Full-Batch | 20% | 12.6(0.1) | 63.9(0.0) | 75.7(0.0) | 67.6(0.0) | 67.0(0.0) | 34.3(0.0) | 50.7(0.0) | 35.4(0.0) | 56.4 |
| Wanda-sp | 20% | 17.4(0.1) | 66.0(0.2) | 75.4(0.1) | 63.0(0.1) | 64.8(0.3) | 33.7(0.0) | 48.2(0.2) | 35.0(0.1) | 55.2 |
| FLAP | 20% | 18.8(0.2) | **68.1(0.4)** | 75.1(0.1) | 62.5(0.2) | 62.6(0.3) | 31.8(0.3) | 49.5(0.1) | 34.5(0.1) | 54.9 |
| PP | 20% | **14.7(0.1)** | 67.4(0.1) | 75.5(0.1) | 65.7(0.0) | 64.9(0.3) | 33.8(0.1) | 51.6(0.0) | 36.5(0.2) | **56.5** |
| Full-Batch | 40% | 17.9(0.2) | 52.1(0.0) | 75.7(0.0) | 64.8(0.0) | 65.5(0.0) | 32.8(0.0) | 50.1(0.0) | 36.8(0.0) | 54 |
| Wanda-sp | 40% | 42.7(0.7) | **63.7(0.1)** | 71.8(0.3) | 53.2(0.1) | 57.6(0.2) | 29.6(0.4) | 43.3(0.1) | 34.3(0.2) | 50.5 |
| FLAP | 40% | 51.0(0.7) | 62.7(0.0) | 72.4(0.0) | 53.3(0.2) | 58.3(0.5) | 29.4(0.3) | 45.2(0.4) | 34.1(0.1) | 50.8 |
| PP | 40% | **26.7(0.3)** | 61.1(0.2) | 74.3(0.1) | 58.7(0.0) | 59.3(0.1) | 33.6(0.1) | 49.7(0.1) | 35.3(0.4) | 53.1 |

Table 17: Zero-shot performance of pruning LLaMA-3-8B with MLP pruned. PP consistently demonstrates superior performance across nearly all tested scenarios.

| Method | Pruning Ratio | WikiText2 | BoolQ | PIQA | HellaSwag | WinoGrande | ARC-c | ARC-e | OBQA | Average |
|---|---|---|---|---|---|---|---|---|---|---|
| Dense | 0% | 6.8(0.0) | 81.7(0.0) | 79.5(0.0) | 76.3(0.0) | 72.5(0.0) | 47.2(0.0) | 61.7(0.0) | 40.2(0.0) | 65.6 |
| Full-Batch | 20% | 8.5(0.0) | 79.0(0.0) | 80.1(0.0) | 74.8(0.0) | 73.9(0.0) | 44.9(0.0) | 60.7(0.0) | 40.2(0.0) | 64.8 |
| Wanda-sp | 20% | 10.0(0.0) | 75.1(0.3) | 78.5(0.0) | 69.6(0.2) | 71.4(0.4) | 38.7(0.4) | 56.9(0.4) | 39.0(0.2) | 61.3 |
| FLAP | 20% | 10.0(0.0) | **79.4(0.2)** | **78.7(0.1)** | 70.3(0.0) | 71.4(0.5) | 40.8(0.1) | 57.8(0.0) | 39.4(0.3) | 62.5 |
| PP | 20% | **9.3(0.0)** | 77.4(0.0) | 78.5(0.0) | 73.1(0.0) | 72.5(0.3) | 43.2(0.3) | 59.1(0.2) | 40.2(0.5) | **63.4** |
| Full-Batch | 40% | 12.3(0.1) | 73.1(0.0) | 77.8(0.0) | 70.5(0.0) | 70.3(0.0) | 42.9(0.0) | 58.9(0.0) | 39.8(0.0) | 61.9 |
| Wanda-sp | 40% | 18.4(0.1) | 66.6(0.1) | 73.4(0.2) | 56.7(0.1) | 63.2(0.2) | 31.8(0.2) | 47.0(0.5) | 34.5(0.2) | 53.3 |
| FLAP | 40% | 18.5(0.2) | 67.3(1.0) | 73.5(0.0) | 57.2(0.2) | 66.7(0.5) | 31.7(0.3) | 44.6(0.3) | 34.4(0.3) | 53.6 |
| PP | 40% | **14.9(0.1)** | 70.3(0.1) | 76.3(0.2) | 65.3(0.1) | 67.2(0.2) | 39.0(0.3) | 57.4(0.1) | 36.9(0.3) | 58.9 |

Table 18: Comparison of PP with fine-tuned baselines on LLaMA-2-7B model, with attention and MLP layers pruned: PP consistently outperforms across scenarios without fine-tuning.

| Method | Pruning Ratio | Fine-tuning | WikiText2 | BoolQ | PIQA | HellaSwag | WinoGrande | ARC-c | ARC-e | OBQA | Average |
|--------|---------------|-------------|-----------|-------|------|-----------|------------|-------|-------|------|---------|
| Dense | 0% | ✗ | 6.0(0.1) | 74.6(0.0) | 77.9(0.0) | 75(0.0) | 67.7(0.0) | 42.7(0.0) | 67.3(0.0) | 42.6(0.0) | 64.0 |
| LoRAPrune w/ LoRA | 20% | ✓ | 8.7(0.2) | 67.0(0.9) | 76.5(0.2) | 69.9(0.1) | 63.2(0.3) | 36.7(0.2) | 58.9(0.9) | 42.3(0.2) | 59.2 |
| LLM-Pruner w/ LoRA | 20% | ✓ | 10.2(0.3) | 66.6(1.3) | 76.1(0.6) | 68.4(0.5) | 62.8(1.1) | 36.3(0.4) | 59.8(0.3) | 40.7(0.7) | 58.7 |
| PP | 20% | ✗ | 8.1(0.1) | 69.0(0.1) | 78.1(0.0) | 73.5(0.0) | 66.7(0.3) | 42.8(0.1) | 68.5(0.0) | 40.9(0.2) | 62.8 |
| LoRAPrune w/ LoRA | 40% | ✓ | 13.6(0.4) | 62.9(0.2) | 70.8(0.1) | 58.6(0.1) | 55.5(0.7) | 30.9(0.4) | 49.6(0.4) | 36.7(0.4) | 52.1 |
| LLM-Pruner w/ LoRA | 40% | ✓ | 20.3(1.3) | 57.5(4.0) | 71.3(1.2) | 55.7(1.3) | 53.1(0.5) | 28.9(0.7) | 50.4(0.5) | 37.3(0.6) | 50.6 |
| PP | 40% | ✗ | 16.8(0.1) | 62.7(0.2) | 74.9(0.1) | 63.6(0.0) | 57.5(0.2) | 35.5(0.1) | 61.7(0.2) | 40.3(0.4) | 56.6 |

Table 19: Comparison of PP with fine-tuned baselines on LLaMA-2-13B model, with attention and MLP layers pruned: PP consistently outperforms across scenarios without fine-tuning.

| Method | Pruning Ratio | Fine-tuning | WikiText2 | BoolQ | PIQA | HellaSwag | WinoGrande | ARC-c | ARC-e | OBQA | Average |
|--------|---------------|-------------|-----------|-------|------|-----------|------------|-------|-------|------|---------|
| Dense | 0% | ✗ | 5.1(0.1) | 72.1(0.0) | 79.6(0.0) | 78.7(0.0) | 70.7(0.0) | 46.5(0.0) | 71.3(0.0) | 44.2(0.0) | 66.2 |
| LoRAPrune w/ LoRA | 20% | ✓ | 7.4(0.0) | 64.4(0.5) | 78.1(0.1) | 74.8(0.2) | 66.0(0.3) | 40.4(0.3) | 61.7(0.9) | 41.6(0.2) | 61.0 |
| LLM-Pruner w/ LoRA | 20% | ✓ | 8.4(0.5) | 70.2(1.4) | 78.3(0.3) | 73.8(0.3) | 65.8(1.3) | 40.1(0.5) | 64.2(0.4) | 42.0(0.4) | 62.1 |
| PP | 20% | ✗ | 6.7(0.1) | 72.0(0.2) | 79.5(0.1) | 77.6(0.0) | 68.5(0.1) | 44.7(0.2) | 71.5(0.1) | 43.0(0.2) | 65.3 |
| LoRAPrune w/ LoRA | 40% | ✓ | 11.1(0.3) | 62.5(0.1) | 74.1(0.4) | 65.5(0.1) | 60.4(0.3) | 33.0(0.4) | 53.9(0.7) | 39.3(0.6) | 55.5 |
| LLM-Pruner w/ LoRA | 40% | ✓ | 15.3(0.7) | 63.9(0.4) | 73.5(0.6) | 62.4(1.4) | 57.5(1.1) | 33.2(1.2) | 55.2(0.7) | 37.5(0.8) | 54.7 |
| PP | 40% | ✗ | 11.3(0.1) | 65.8(0.1) | 77.1(0.2) | 71.6(0.0) | 61.3(0.4) | 40.9(0.3) | 67.9(0.1) | 42.5(0.3) | 61.0 |

