# OpenReview forum: "Probe Pruning: Accelerating LLMs through Dynamic Pruning via Model-Probing"
_ICLR.cc/2025/Conference — ICLR 2025 Poster_

### Official Review · Reviewer_vvRG · 2024-10-22

**Soundness:** 3
**Presentation:** 3
**Contribution:** 3
**Rating:** 8
**Confidence:** 3

**Summary:**

The paper acknowledges challenges incurred by the majority of pruning methods:
* That relying on a calibration dataset to collect pruning statistics introduces potential bias, in that the calibration dataset cannot be perfectly representative of real-world data.
* That pruning requires access to information, such as gradients, that are not usually available during inference, precluding dynamic pruning.

The paper hints at the fact that dynamic pruning is poised to surpass dynamic pruning in performance and proposes a framework to make it practically feasible.

In the paper, the authors recognize that structured pruning is, according to most implementations, the only feasible way to effectively speed up inference.

The paper introduces probing, as a means for collecting statistics required for pruning at negligible additional cost.

The paper provides a configuration named Full-Batch Pruning that gives an upper-bound limit to the accuracy of dynamic pruning. This can be used to better measure the effect of sampling during probing. The paper shows that the selection of prunable weights is more similar to that of Full-Batch Pruning for PP than it is with the fixed-prune models.

History-informed pruning is introduced the alleviate the inaccuracy of probing when the sampling rate is small, by accumulating statistics over multiple batches.

A pruning metric is introduced. The metric is based on the Wanda metric, with an additional squaring to capture the importance of individual weights.

The paper concludes with a large amount of empirical results and detailed ablations in the appendix.

**Strengths:**

The challenges described in the introduction are indeed fundamental limitations of most existing pruning methods.
Dynamic Pruning is a very promising research avenue and this paper brings several novel aspects to it: probing, pruning metric.

The paper provides a good set of baselines to compare against.

The paper validates the method with several model backbones (LLaMA2, LLaMA3.1, OPT) and benchmarks that include text generation and common reasoning tasks.

Every design choice in the paper is carefully described, ablated and proven with experimental results.

**Weaknesses:**

The method relies on having rather large batch sizes and sequence lengths to sample from in order to generate a probe. However in many inference setups where speed matters and thus pruning is desirable, the batch size is small, sometimes down to 1. Similarly, inference time in LLMs tends to be dominated by the auto-regressive (aka generation) phase, where tokens are produced one at a time.

The inference measurements and FLOPs calculations demonstrate the small computational complexity of running the probe. However in practice, I understand that we need to run the probe first and only then do we know which filters to prune. Potentially, the probes can run in parallel with the actual computation of earlier pruned blocks. In order to prove the point that the benefits of probing + dynamic pruning outweigh the cost of probing, it would be nice to have end-to-end inference measurements.

The paper does not compare against the Minitron baseline (https://arxiv.org/pdf/2408.11796), yet this paper shows better metrics. For example for LLaMA3.1-4B, LLaMA3.1-Minitron4B(50% pruning) has HellaSwag at 76.1, while PP has it at 65.3 with 40% pruning. Thus it is unclear whether the method beats the State of the Art.

There is empirical data to probe the superiority of the proposed pruning metric but no theoretical justification.

**Questions:**

1. Will history-informed pruning counterbalance the limited sampling possibilities when the batch size of sequence length are small?

2. Can you run end-to-end inference measurements with PP in order to show how much speed up we can expect from 20% and 40% pruning ratios?

3. Does PP speed up the auto-regressive phase of inference?

4. Can you compare against the Minitron baseline?

5. Can you further explain why the proposed pruning metric is superior to the Wanda metric?

**Details Of Ethics Concerns:**

None.

---

> ### Author Response · Authors · 2024-11-21
> **Response**
>
> We sincerely appreciate the reviewer's insightful feedback and positive comments on the methodology of our paper. Please find our responses below:
>
> **Response to Weakness 1, Question 1, and Question 3:**
>
> We appreciate the reviewer's insightful feedback. We address Weakness 1, Question 1, and Question 3 here, as these three questions are closely related.
>
> First, we would like to discuss what happens when the batch size is small and how history-informed pruning affects the final performance when the sampling sequence length is small. Note that when generating probes, PP prunes both the sequence and batch dimensions by selecting only key samples and tokens. When the batch size is small, PP can reduce the probe's sequence dimension and still maintain comparable results (or even better), as we perform online structured pruning in a batch-wise manner. If there are not many samples, we can reduce the probe's sequence dimension and still achieve similar performance.
>
> We conducted an experiment on LLaMA-2-7B where we decreased the batch size to 5 for both tasks (in the paper, we used a batch size of 20). Moreover, we set the default probe size to 1 sample and 15\% of the sequence length (we used 50\% in the paper), resulting in approximately 1.5\% FLOPs cost. We refer to this scenario as **PP-Limit**. Furthermore, we conducted an experiment on PP without using history-informed pruning, denoted as **PP-Limit w/o History-Informed Pruning**, while **PP-Limit w/ History-Informed Pruning** indicates that we still use history-informed pruning. In the table, **PP** represents the default setting used in the main text of the paper.
>
> From Table 1, we observe that with history-informed pruning, PP under the limited scenario can obtain results comparable to PP's default setting. For example, at a 40\% pruning ratio, PP-Limit w/ History-Informed Pruning achieves a perplexity of 16.8 on WikiText2 and 62.4\% accuracy on ARC-e, which is comparable to PP (or even better), which has a perplexity of 16.8 on WikiText2 and 61.7\% accuracy on ARC-e. We also observe that without history-informed pruning, the performance is not stable and is worse than with history-informed pruning. For instance, at a 40\% pruning ratio, PP-Limit w/o History-Informed Pruning achieves a perplexity of 21.5 on WikiText2 and 50.0\% accuracy on ARC-e. Although it performs better than baselines such as FLAP (which has a perplexity of 38.9 on WikiText2) on WikiText2, its accuracy on ARC-e is lower than FLAP's 52.5\%.
>
> From these results, we can conclude that PP can handle the small batch size case, as we can reduce the probe's sequence dimension and still maintain comparable performance. Furthermore, history-informed pruning helps improve performance when the sampling sequence length is small, as it allows PP to leverage historical information.
>
> Furthermore, our primary focus is online dynamic structured pruning during the prefilling stage, where the model processes hundreds of tokens before generating the first one. We leave the exploration of probe pruning for the online dynamic structured pruning during the decoding stage to future work.
>
> | Method                                | Pruning Ratio | WikiText2 &darr;| ARC-e &uarr; |
> |---------------------------------------|---------------|------------|----------|
> | Dense                                 | 0\%            | 6.0(0.1)   | 67.3(0.0)|
> | Full-Batch Probing                    | 20\%           | 7.3(0.1)   | 67.2(0.0)|
> | Wanda-sp                              | 20\%           | 10.6(0.1)  | 63.9(0.3)|
> | FLAP                                  | 20\%           | 10.3(0.1)  | 63.1(0.1)|
> | PP-Limit w/o History-Informed Pruning | 20\%           | 8.6(0.1)   | 63.8(0.0)|
> | PP-Limit w/ History-Informed Pruning  | 20\%           | 8.1(0.1)   | 68.8(0.1)|
> | PP                                    | 20\%           | 8.1(0.1)   | 68.5(0.0)|
> | Full-Batch Probing                    | 40\%           | 13.6(0.1)  | 64.7(0.0)|
> | Wanda-sp                              | 40\%           | 43.8(1.5)  | 54.4(0.1)|
> | FLAP                                  | 40\%           | 38.9(1.3)  | 52.5(0.2)|
> | PP-Limit w/o History-Informed Pruning | 40\%           | 21.5(0.2)  | 50.0(0.0)|
> | PP-Limit w/ History-Informed Pruning  | 40\%           | 16.8(0.2)  | 62.4(0.2)|
> | PP                                    | 40\%           | 16.8(0.1)  | 61.7(0.2)|
>
> *Table 1: Zero-shot performance of LLaMA-2-7B after pruning attention and MLP blocks without
> fine-tuning.*

---

> > ### Author Response · Authors · 2024-11-21
> > **Response continued**
> >
> > **Response to Weakness 2 and Question 2:**
> >
> > We thank the reviewer for the insightful comment.
> >
> > First, we verify the possibility of running the probe in parallel with the actual computation of earlier pruned blocks. We present the results in Table 2 below. Here, **PP-Parallel** represents the approach where, when the actual computation happens on earlier pruned blocks, we generate the probe from the residuals of these earlier pruned blocks and perform the probing. **PP** represents the default setting used in the main text of the paper. The results show that we can still obtain performance gains and achieve comparable results to PP. For example, at a 40\% pruning ratio, PP-Parallel achieves a perplexity of 17.9 on WikiText2, which is close to that of PP and much lower than the 38.9 achieved by FLAP. Furthermore, PP-Parallel achieves 61.4\% accuracy on ARC-e, which is close to that of PP and much higher than the 52.5\% achieved by FLAP. However, we are just demonstrating the feasibility of further improving PP's inference speed here; the actual parallelism is hardware-dependent and implementation-dependent.
> >
> > Second, we perform end-to-end runtime measurements with PP and present the results in Table 3. From Table 3, we can see that PP obtains the best Performance Runtime Ratio, which means PP experiences lowest performance degradation per unit of runtime reduction among three methods. On the other hand, the runtime of PP may be slightly lower than those of other structured pruning baselines; however, the end-to-end runtime is highly hardware-dependent and implementation-dependent. We believe that the inference speed of PP could be improved through implementation optimizations or other techniques, and we leave this to future work.
> >
> > Furthermore, we fix a typo in Table 5 of the main text, as the results in Table 5 are measured across all batches of WikiText2. Please find our edited PDF.
> >
> > | Method            | Pruning Ratio | WikiText2 &darr;| ARC-e &uarr;    |
> > |-------------------|---------------|-----------|-----------|
> > | Dense             | 0\%            | 6.0(0.1)  | 67.3(0.0) |
> > | Full-Batch Probing| 20\%           | 7.3(0.1)  | 67.2(0.0) |
> > | Wanda-sp          | 20\%           | 10.6(0.1) | 63.9(0.3) |
> > | FLAP              | 20\%           | 10.3(0.1) | 63.1(0.1) |
> > | PP-Parallel       | 20\%           | 8.1(0.1)  | 67.9(0.1) |
> > | PP                | 20\%           | 8.1(0.1)  | 68.5(0.0) |
> > | Full-Batch Probing| 40\%           | 13.6(0.1) | 64.7(0.0) |
> > | Wanda-sp          | 40\%           | 43.8(1.5) | 54.4(0.1) |
> > | FLAP              | 40\%           | 38.9(1.3) | 52.5(0.2) |
> > | PP-Parallel       | 20\%           | 17.9(0.1) | 61.4(0.2) |
> > | PP                | 40\%           | 16.8(0.1) | 61.7(0.2) |
> >
> > *Table 2: Zero-shot performance of LLaMA-2-7B after pruning attention and MLP blocks without fine-tuning.*
> >
> > | Method   | Pruning Ratio | Performance Runtime Ratio &darr; | Runtime(s)   | Speedup    |
> > |----------|---------------|------|--------------|------------|
> > | Dense    | 0\%            | -    | 31.69(0.16)  | -          |
> > | Wanda-sp | 20\%           | 0.85 | 26.28(0.12)  | 1.21×      |
> > | FLAP     | 20\%           | 0.85 | 26.63(0.10)  | 1.19×      |
> > | PP       | 20\%           | 0.60  | 28.21(0.03)  | 1.12×      |
> > | Wanda-sp | 40\%           | 2.71 | 20.57(0.05)  | 1.54×      |
> > | FLAP     | 40\%           | 2.42 | 21.26(0.07)  | 1.49×      |
> > | PP       | 40\%           | 1.16 | 22.40(0.13)  | 1.41×      |
> >
> > *Table 3: Performance runtime ratio and end-to-end runtime across all batches of WikiText2.*

---

> > > ### Author Response · Authors · 2024-11-21
> > > **Response continued**
> > >
> > > **Response to Weakness 3 and Question 4:**
> > >
> > > We would like to thank the reviewer for letting us know this interesting paper. After reviewing it, we believe that comparing our method against Minitron may be less appropriate than comparing to the most relevant baselines in our paper, such as FLAP and Wanda-sp. We provide two reasons:
> > >
> > > 1) Minitron requires an extensive full-parameter iterative fine-tuning procedure, whereas PP, along with FLAP and Wanda-sp, is based on inference without the need for fine-tuning. Furthermore, Minitron incorporates QA data [1] into the pruning and distillation processes (as detailed in the Minitron Training Details section), whereas PP only uses the C4 dataset for calibration. We believe that the inclusion of QA data could help improve the HellaSwag performance, as it is also a QA dataset (as shown in Table 13 of [1]). Minitron first fine-tunes the teacher model (LLaMA3.1-8B) using their own 127B tokens. This is followed by iterative pruning and distillation for the student model using another 94B tokens. It's important to note that full-parameter fine-tuning is much more resource-intensive than parameter-efficient fine-tuning, and distillation incurs additional computational costs due to the need for running a teacher model.
> > >
> > > 2) Minitron conducts the HellaSwag experiment under a few-shot setting, as outlined in Table 1 of the Minitron paper, whereas PP conducts the HellaSwag experiment under a zero-shot setting. In the zero-shot setting, PP does not provide task-specific examples during inference to guide the model's responses. Conversely, in the few-shot setting, the model is prompted with task-specific examples [2, 3]. For instance, a few-shot setting might use a prompt such as "Examples: 3+5=8. 2+3=5. Question: What is the result for 4+6=?". However, PP does not employ such prompts during inference. This variation in prompting during evaluation may account for Minitron's superior performance on the HellaSwag dataset.
> > >
> > > [1] Parmar, Jupinder, et al. "Reuse, Don't Retrain: A Recipe for Continued Pretraining of Language Models." arXiv preprint arXiv:2407.07263 (2024).
> > >
> > > [2] Wei, Jason, et al. "Finetuned language models are zero-shot learners." arXiv preprint arXiv:2109.01652 (2021).
> > >
> > > [3] Brown, Tom B. "Language models are few-shot learners." arXiv preprint arXiv:2005.14165 (2020).
> > >
> > > **Response to Weakness 4 and Question 5:**
> > >
> > > Our proposed PPsp metric has superior performance to the Wanda-sp metric (Wanda metric for structured pruning; both Wanda-sp and Wanda metric use the same importance score of individual weights) due to two main differences. First, we square the Wanda-sp metric importance score of individual weights. Second, we calculate the L2 norm of the importance scores across pruning structures, whereas Wanda-sp sums the importance scores across pruning structures.
> > >
> > > We explain the first difference here. Wanda's metric is derived from SparseGPT's metric and initially includes a squaring term. However, Wanda discards this squaring (as shown in Equation 1 of Wanda's paper and in their official implementation: https://github.com/locuslab/wanda/blob/main/lib/prune.py) because Wanda performs unstructured pruning. Unstructured pruning relies on ranking importance scores on a per-output basis, so removing the squaring term does not affect the relative ranking of importance scores in Wanda's approach. In contrast, for structured pruning, retaining the squaring of individual weights' importance scores is essential. This is because structured pruning involves removing entire model structures, and maintaining the squared importance scores ensures that the inherent importance of each weight is accurately preserved.
> > >
> > > For the second difference, we empirically demonstrate that using the L2 norm of importance scores across pruning structures leads to better performance than summing the importance scores across pruning structures. This effectiveness is supported by our experimental results, as shown in Table 7 of the main text.
> > >
> > > We agree that adding a theoretical justification for the proposed pruning metric would enhance its credibility. We thank the reviewer for this valuable suggestion.

---

> > > > ### Author Response · Authors · 2024-11-25
> > > > **A kind reminder**
> > > >
> > > > Dear Reviewer vvRG,
> > > >
> > > > We would like to thank you again for reviewing our paper and providing valuable, positive comments. We believe that we have addressed your concerns. Since the Author/Reviewer Discussion Stage is ending soon and we have not heard back from you yet, we would appreciate if you could kindly let us know of any other concerns you might have, and if we can be of any further assistance in clarifying any issues.
> > > >
> > > > Best regards,
> > > >
> > > > Authors

---

> > > > > ### Comment · Reviewer_vvRG · 2024-11-30
> > > > > **Response to feedback**
> > > > >
> > > > > Hello, I would like to thank the authors for the detailed responses. The responses have addressed by questions. In particular, I note how history-informed probe pruning helps compensate for the case when the batch size is small.
> > > > >
> > > > > In light of the responses, I would like to keep my score (Accept). Thank you!

---

> > > > > > ### Author Response · Authors · 2024-12-01
> > > > > > **Thank you again!**
> > > > > >
> > > > > > Dear Reviewer vvRG,
> > > > > >
> > > > > > Thank you once again for reviewing our paper and providing valuable, positive comments. They have been extremely helpful in enhancing our work.
> > > > > >
> > > > > > Best regards,
> > > > > >
> > > > > > Authors

---

> > ### Comment · Area_Chair_mBUN · 2024-11-30
> >
> > Dear Reviewer,
> >
> > Could you kindly respond and indicate whether authors have addressed your concerns?
> >
> > Thanks, AC

---

### Official Review · Reviewer_NzRF · 2024-11-01

**Soundness:** 3
**Presentation:** 2
**Contribution:** 4
**Rating:** 8
**Confidence:** 5

**Summary:**

The paper proposes an approach for on the fly pruning of LLM. On the fly means that the structure is changing depending on the prompt. The approach is motivated by observation that using different data for calibration results in different pruning structures. To make the approach more efficient, only a portion of samples and tokens is selected for importance estimation. Results are solid and show a trade-off between improvements in accuracy, but also a slowdown in throughput.

**Strengths:**

- The problem is interesting and novel.
- The approach is well motivated and look novel.
- The motivation behind selective importance estimation is great.
- I like the way the importance is tracked and updated from initial states.
- Results are solid, they show clear benefits of the pruning "on the fly"

**Weaknesses:**

- The improvement in perplexity comes with reduction in model throughput as shown in Table 5.
- The method has weaknesses when compared to other techniques. For example the entire model needs to be load, whereas for other structured pruning techniques, only the pruned version is required. Limitations and weaknesses are not highlighted in Table 1 forming an incomplete story.
- The paper is written in a complicated way, it feels that the idea and method can be described in a more intuitive and simpler way. I don't count this as a strong weakness, but rather encourage authors to improve the paper for future readers.
- For evaluations, is it implemented as the entire batch consists of a single task? What happened when the batch is mixed? I would assume degradation of the current approach.

**Questions:**

Statements as on page 1 need empirical evidence to backup the statement: "Although the calibration dataset provides valuable insights for pruning by identifying non-critical weights, this approach overlooks the batch-dependent nature of outlier properties in LLMs Liu et al. (2023b); Song et al. (2023); Liu et al. (2024); Sun et al."

Paper talks about "residual importance" a lot, but this concept is not clear before page 5. It would be better to introduce it earlier. Does it mean an L2 norm of the hidden state? If yes, then it will be better to simplify description.

How to understand "Full-Batch Probing" - is it just a standard structured pruning? How is that different from other techniques?

From tables 2-3 it seems that full-batch probing outperforms all other techniques, is it because pruning mask changes between batches?

What is pruned exactly? Are heads pruned, in intermediate dimension in MLP pruned, what about hidden dimension and layers?

---

> ### Author Response · Authors · 2024-11-21
> **Response**
>
> We sincerely appreciate the reviewer's insightful feedback and positive comments on the methodology of our paper. Please find our responses below:
>
> **Response to Weakness 1:**
>
> In Table 5, we show a trade-off between performance improvements and inference speed. We demonstrate that PP achieves the best Performance Runtime Ratio, which quantifies the ratio of performance degradation per unit of runtime reduction, in comparison with FLAP and Wanda-sp. Specifically, PP's Performance Runtime Ratio values are 2.56× (95.65 compared to 37.37) and 2.85× (106.48 compared to 37.37) more efficient than those of FLAP and Wanda-sp, respectively, indicating a significantly lower rate of performance degradation. Conversely, regarding final speedups, the inference speeds of PP are slightly slower but comparable to other structured pruning baselines, such as FLAP and Wanda-sp. It is important to note that inference speed is highly hardware-dependent and implementation-dependent. We believe that the inference speed of PP could be enhanced through further implementation optimizations or other techniques, which we plan to explore in future research.
>
> Furthermore, we have explored the feasibility of running the probe in parallel with the actual computation of earlier pruned blocks. We present these results in Table 1. Here, **PP-Parallel** represents the approach where, when the actual computation happens on earlier pruned blocks, we generate the probe from the residuals of these earlier pruned blocks and perform the probing. **PP** represents the default setting used in the main text of the paper. The results show that we can still obtain performance gains and achieve comparable results to PP. For example, at a 40\% pruning ratio, PP-Parallel achieves a perplexity of 17.9 on WikiText2, which is close to that of PP and much lower than the 38.9 achieved by FLAP. Furthermore, PP-Parallel achieves 61.4\% accuracy on ARC-e, which is close to that of PP and much higher than the 52.5\% achieved by FLAP. However, we are just demonstrating the feasibility of further improving PP's inference speed here; the actual parallelism is hardware-dependent and implementation-dependent.
>
> | Method            | Pruning Ratio | WikiText2 &darr;| ARC-e &uarr;    |
> |-------------------|---------------|-----------|-----------|
> | Dense             | 0\%            | 6.0(0.1)  | 67.3(0.0) |
> | Full-Batch Probing| 20\%           | 7.3(0.1)  | 67.2(0.0) |
> | Wanda-sp          | 20\%           | 10.6(0.1) | 63.9(0.3) |
> | FLAP              | 20\%           | 10.3(0.1) | 63.1(0.1) |
> | PP-Parallel       | 20\%           | 8.1(0.1)  | 67.9(0.1) |
> | PP                | 20\%           | 8.1(0.1)  | 68.5(0.0) |
> | Full-Batch Probing| 40\%           | 13.6(0.1) | 64.7(0.0) |
> | Wanda-sp          | 40\%           | 43.8(1.5) | 54.4(0.1) |
> | FLAP              | 40\%           | 38.9(1.3) | 52.5(0.2) |
> | PP-Parallel       | 20\%           | 17.9(0.1) | 61.4(0.2) |
> | PP                | 40\%           | 16.8(0.1) | 61.7(0.2) |
>
> *Table 1: Zero-shot performance of LLaMA-2-7B after pruning attention and MLP blocks without fine-tuning.*
>
> **Response to Weakness 2:**
>
> We sincerely appreciate the reviewer's insightful feedback. We have added the sentence "Our implementation loads the full model for dynamic pruning, while other methods load only the pruned version." to the caption of Table 1. The revised sentences is highlighted in red in our edited PDF.
>
> **Response to Weakness 3:**
>
> We sincerely thank the reviewer for the valuable feedback. We will revise the paper accordingly to improve its readability for future readers.

---

> > ### Author Response · Authors · 2024-11-21
> > **Response continued**
> >
> > **Response to Weakness 4:**
> >
> > We suppose the "batch is mixed" mentioned by the reviewer means that each batch consists of two tasks: commonsense reasoning task and text generation task. We conduct an experiment to verify the "batch is mixed" scenario. Within each batch, we allocate the first 50\% of the samples to the commonsense reasoning task (ARC-e) and the remaining 50\% to the text generation task (WikiText2). We would like to respectfully note that due to the misalignment of dataset-dependent properties, such as sample length and the number of samples in the datasets (WikiText2 and ARC-e), we have attempted to minimize these discrepancies, but it might not be the ideal comparison case. We set the batch size to 20 and the sequence length to 128, selecting this length because the samples in ARC-e have varying lengths, ranging from approximately 10–20 tokens up to 200 tokens, with most samples being within 128 tokens. We sample additional sentences from WikiText2 to align the number of samples with those of ARC-e. We set the default probe size to 5\% of the batch size and 50\% of the sequence length, consistent with our experimental settings.
> >
> > The results are presented in the Table 2. **PP-Mix** denotes the scenario where two tasks are included in one batch. **PP** represents the default setting used in the main text of the paper. We observe that PP-Mix can still maintain performance improvements compared to baselines, which might be due to the hidden shared linguistic structures between samples. These shared structures can enable our method to effectively identify and preserve important weights that are beneficial for both tasks. Compared to PP, we find that PP-Mix has fluctuating performance in two tasks (better on WikiText2, worse on ARC-e). We believe this might be caused by the aforementioned misalignment of dataset-dependent properties. For instance, at a 20\% pruning ratio, compared to PP, PP-Mix's perplexity on WikiText2 decreased from 16.6 to 16.1, and the accuracy on ARC-e dropped from 68.5\% to 67.1\%. At a 40\% pruning ratio, compared to PP, PP-Mix's perplexity on WikiText2 decreased from 33.2 to 30.7, and the accuracy on ARC-e dropped from 61.7\% to 55.3\%.
> >
> > | Method            | Pruning Ratio | WikiText2 &darr;  | ARC-e &uarr;  |
> > |-------------------|---------------|--------------|----------|
> > | Dense             | 0\%            | 12.1(0.0)    | 67.3(0.0)|
> > | Full-Batch Probing| 20\%           | 15.2(0.0)    | 67.2(0.0)|
> > | Wanda-sp          | 20\%           | 30.2(0.1)    | 63.9(0.3)|
> > | FLAP              | 20\%           | 27.7(0.3)    | 63.1(0.1)|
> > | PP-Mix          | 20\%           | 16.1(0.0)    | 67.1(0.0)|
> > | PP                | 20\%           | 16.6(0.1)    | 68.5(0.0)|
> > | Full-Batch Probing| 40\%           | 25.2(0.0)    | 64.7(0.0)|
> > | Wanda-sp          | 40\%           | 68.2(0.6)    | 54.4(0.1)|
> > | FLAP              | 40\%           | 65.5(0.8)    | 52.5(0.2)|
> > | PP-Mix          | 40\%           | 30.7(0.0)    | 55.3(0.6)|
> > | PP                | 40\%           | 33.2(0.1)    | 61.7(0.2)|
> >
> > *Table 2: Zero-shot performance of LLaMA-2-7B after pruning attention and MLP blocks without
> > fine-tuning.*
> >
> > **Response to Question 1:**
> >
> > We thank the reviewer for the insightful feedback. Due to the page limit of the main text, we have added a reference sentence in the introduction (marked in red) and included a new section (also marked in red) titled "Batch-Dependent Outliers at Token Positions" in Appendix B.3. We present the calculated L2 norms for each token position of the input hidden states at layers 10 and 20 across the batch and feature dimensions in Figure 5 of the Appendix. The results demonstrate the presence of batch-dependent outliers at each token position, aligning with the observations from existing works [1, 2].
> >
> > [1] Liu, Ruikang, et al. "IntactKV: Improving Large Language Model Quantization by Keeping Pivot Tokens Intact." arXiv preprint arXiv:2403.01241 (2024).
> >
> > [2] Sun, Mingjie, et al. "Massive activations in large language models." arXiv preprint arXiv:2402.17762 (2024).
> >
> > **Response to Question 2:**
> >
> > Yes, the residual importance is the L2 norm of the input hidden states across specific dimensions, as shown in the Equation 4 and 5 of the main text. We introduce the term "residual importance" to distinctly define this concept and differentiate input hidden states from layernormalized hidden states, intermediate hidden states, and output hidden states for clearer understanding.
> >
> > We sincerely appreciate the writing suggestion and agree that the introduction of this concept should be both earlier and more explicit. We have now introduced residual importance in Section 'Notations and Preliminaries' on page 3, and have revised sentences in the 'Probe Generation' section on page 4 and 5. The revised sentences are highlighted in red in our edited PDF.

---

> > > ### Author Response · Authors · 2024-11-21
> > > **Response continued**
> > >
> > > **Response to Question 3 and Question 4:**
> > >
> > > We regard Full-Batch Probing as the performance upper bound of our method. Recall that in PP, when generating probes, PP prunes both the sequence and batch dimensions by selecting only key samples and tokens. In our experimental setting, we utilize 5\% of the batch size and 50\% of the sequence length to form the probe. However, in Full-Batch Probing, we utilize 100\% of the batch size and 100\% of the sequence length, encompassing all input hidden states, to process a few model layers. As such, Full-Batch Probing captures the complete intermediate hidden state information, potentially leading to the best dynamic pruning performance. However, this approach significantly increases computational resource requirements and latency. For example, at a 40\% pruning ratio, the extra computational cost for Full-Batch Probing is 60\% (compared to PP's 1.5\%), which exceeds our pruning goals and is impractical in real application scenarios. Thus, we consider it as a performance upper bound of our method. The aim of probing is to select pruning channels similar to those chosen by Full-Batch Probing and to approach Full-Batch Probing's performance with significantly lower computational costs.
> > >
> > > **Response to Question 5:**
> > >
> > > Currently, our approach focuses on pruning attention heads and the intermediate dimensions in the MLP layers. However, we agree that probing techniques may enhance our ability to prune hidden dimensions and layers as well. For instance, we could generate a probe from the current batch and use it to assess the entire next layer to determine if we can skip that layer entirely for the current batch. We acknowledge these unexplored directions and leave them for future work.

---

> > > > ### Author Response · Authors · 2024-11-25
> > > > **A kind reminder**
> > > >
> > > > Dear Reviewer NzRF,
> > > >
> > > > We would like to thank you again for reviewing our paper and providing valuable, positive comments. We believe that we have addressed your concerns. Since the Author/Reviewer Discussion Stage is ending soon and we have not heard back from you yet, we would appreciate if you could kindly let us know of any other concerns you might have, and if we can be of any further assistance in clarifying any issues.
> > > >
> > > > Best regards,
> > > >
> > > > Authors

---

> ### Comment · Area_Chair_mBUN · 2024-11-30
>
> Dear Reviewer,
>
> Could you kindly respond and indicate whether authors have addressed your concerns?
>
> Thanks, AC

---

### Official Review · Reviewer_ebT5 · 2024-11-03

**Soundness:** 3
**Presentation:** 2
**Contribution:** 2
**Rating:** 6
**Confidence:** 4

**Summary:**

This paper presents a  framework for online dynamic structured pruning of Large Language Models (LLMs). The proposed method, called Probe Pruning (PP), utilizes a small subset of hidden states to identify crucial weights, enabling tailored pruning for different batches during inference. Key components of the framework include probing, which gathers important intermediate hidden state information, and history-informed pruning, which integrates probing states with historical data to enhance pruning decisions. The results demonstrate that PP significantly improves efficiency and performance compared to existing methods, achieving better outcomes without the need for fine-tuning or additional neural network modules. The paper highlights the potential of PP for optimizing LLMs while maintaining their effectiveness.

**Strengths:**

The paper demonstrates that PP can achieve substantial reductions in computational overhead while maintaining or even improving model performance. The ability to perform dynamic pruning during inference allows for tailored optimizations that adapt to the specific characteristics of each batch, enhancing overall efficiency.


The authors provide comprehensive experimental results across various tasks and models, including LLaMA-2 and OPT. The consistent outperformance of PP compared to existing baselines, including those that involve fine-tuning, underscores the effectiveness and reliability of the proposed method.

**Weaknesses:**

1. Unfair Comparison: This approach is a dynamic inference technique, but it is mainly compared to some static pruning techniques. The static pruning work is generally less accurate than dynamic pruning, but it reduces the LLM storage overhead and accelerates the results consistently. Dynamic pruning cannot reduce the LLM storage overhead and the acceleration result varies greatly under different inputs.

2. Lack of comparison of other dynamic inference approaches. For example, some dynamic early exits can be seen as dynamic depth pruning, but also other token-level dynamic decoding methods. Authors should present comprehensive comparisons and discussions, and clearly explain the advantages of dynamic weight pruning over other dynamic inference methods.

3. Not enough impressive novelty: The Historical States method is common operation similar to  EMA or Memory Bank in contrast learning. I consider this technique to be a trade-off strategy rather than an essential solution.


4. Absence of recent LLM evaluations: this work mainly evaluates LLMs released in 2023, the latest LLMs released in 2024 are lacking in experiments. e.g., Llama 3 70B,  Llama 3.2 1B and 3B, DeepSeekMoE，Mixtral-8x7B.

5. Lacking some discussion of recent LLM pruning efforts, such as Pruner-Zero: Evolving Symbolic Pruning Metric From Scratch for Large Language Models  (ICML-2024)  Discovering Sparsity Allocation for Layer-wise Pruning of Large Language Models (NeurIPS-2024) Adaptive Layer Sparsity for Large Language Models via Activation Correlation Assessment  (NeurIPS-2024)

**Questions:**

See Weaknesses

---

> ### Author Response · Authors · 2024-11-21
> **Response**
>
> We sincerely appreciate the reviewer's insightful feedback on the methodology of our paper. Please find our responses below:
>
> **Response to Weakness 1 and Weakness 2:**
>
> We greatly appreciate the reviewer’s feedback and would find it helpful if the reviewer could provide specific paper references for the research areas mentioned, as this would allow us to make more focused and meaningful comparisons.
>
> We compare our method to static pruning techniques because, to the best of our knowledge, there are no existing dynamic pruning methods for the pre-filling stage. Our work is the first to present such a method. Among the methods we identified, static pruning techniques are the closest, as they also perform pruning during the pre-filling stage. However, our approach represents a novel direction by conducting pruning during inference rather than after calibration, thereby offering an additional advantage over static calibration-based methods.
>
> Regarding dynamic depth pruning methods and token-level dynamic decoding methods, we would like to clarify the fundamental distinctions between our method and existing work in these two research areas. Our method primarily focuses on online dynamic pruning during the prefilling stage, wheras main focus of most papers in dynamic depth pruning and token-level dynamic decoding areas is decoding stage.
>
> We acknowledge that static pruning method can reduces the LLM storage overhead while dynamic pruning cannot. We have added the sentence "Our implementation loads the full model for dynamic pruning, while other methods load only the pruned version." to the caption of Table 1. The revised sentences is highlighted in red in our edited PDF.
>
> In this work, we focus on the results concerning performance and acceleration. Regarding performance, we demonstrate that our method consistently outperforms all static pruning baselines across various models and pruning ratios (Tables 2, 3, and 6 in the main text), and we also demonstrate that PP achieves the best Performance Runtime Ratio, which quantifies the ratio of performance degradation per unit of runtime reduction, in comparison with FLAP and Wanda-sp (Table 5 in the main text). In terms of acceleration, we show that the inference speeds of PP are comparable to those of other static structured pruning baselines (Table 5 in the main text). It is important to note that inference speed is highly hardware-dependent and implementation-dependent. We believe that the inference speed of PP could be enhanced through further implementation optimizations or other techniques; we intend to explore these possibilities in future work. For example, we have explored the feasibility of running the probe in parallel with the actual computation of earlier pruned blocks. We present these results in Table 1. Here, **PP-Parallel** represents the approach where, when the actual computation happens on earlier pruned blocks, we generate the probe from the residuals of these earlier pruned blocks and perform the probing. **PP** represents the default setting used in the main text of the paper. The results show that we can still obtain performance gains and achieve comparable results to PP. For example, at a 40\% pruning ratio, PP-Parallel achieves a perplexity of 17.9 on WikiText2, which is close to that of PP and much lower than the 38.9 achieved by FLAP. Furthermore, PP-Parallel achieves 61.4\% accuracy on ARC-e, which is close to that of PP and much higher than the 52.5\% achieved by FLAP. However, we are just demonstrating the feasibility of further improving PP's inference speed here; the actual parallelism is hardware-dependent and implementation-dependent.
>
> | Method            | Pruning Ratio | WikiText2 &darr;| ARC-e &uarr;    |
> |-------------------|---------------|-----------|-----------|
> | Dense             | 0\%            | 6.0(0.1)  | 67.3(0.0) |
> | Full-Batch Probing| 20\%           | 7.3(0.1)  | 67.2(0.0) |
> | Wanda-sp          | 20\%           | 10.6(0.1) | 63.9(0.3) |
> | FLAP              | 20\%           | 10.3(0.1) | 63.1(0.1) |
> | PP-Parallel       | 20\%           | 8.1(0.1)  | 67.9(0.1) |
> | PP                | 20\%           | 8.1(0.1)  | 68.5(0.0) |
> | Full-Batch Probing| 40\%           | 13.6(0.1) | 64.7(0.0) |
> | Wanda-sp          | 40\%           | 43.8(1.5) | 54.4(0.1) |
> | FLAP              | 40\%           | 38.9(1.3) | 52.5(0.2) |
> | PP-Parallel       | 20\%           | 17.9(0.1) | 61.4(0.2) |
> | PP                | 40\%           | 16.8(0.1) | 61.7(0.2) |
>
> *Table 1: Zero-shot performance of LLaMA-2-7B after pruning attention and MLP blocks without fine-tuning.*

---

> ### Author Response · Authors · 2024-11-21
> **Response continued**
>
> **Response to Weakness 3:**
>
> We would like to respectfully point out that the reviewer may have overlooked the main novelty of our paper, which we will clarify here. Our key novelties are primarily presented in two parts: (1) Probing. We leverage the insight that not all samples and tokens contribute equally to the model’s output, and probing a small portion of each batch effectively identifies crucial weights, enabling tailored dynamic pruning for different batches. (2) History-informed pruning with importance-scaled fusion. We strategically merge the probing states with historical states using importance-scaled fusion to capture the essential characteristics of each batch, enhancing the pruning decision.
>
> Regarding the second point, our key merit lies in the strategic combination of probing states with historical states, which captures the essential characteristics of each batch and improves pruning decisions utilizing historical information. The Exponential Moving Average (EMA) is merely a method to update historical states; the reviewer may choose to substitute it with another method.
>
> Furthermore, we have not mentioned contrastive learning at all in the paper, and our paper is irrelevant to contrastive learning. The Memory Bank that the reviewer refers to is fundamentally different from our history-informed pruning. The samples from the Memory Bank are used for calculating contrastive loss in models applying contrastive learning, whereas we utilize importance-scaled fusion to integrate historical states and probing states to guide the pruning decision.
>
> **Response to Weakness 4:**
>
> We would like to respectfully clarify that our research encompasses evaluations of LLMs released in 2024. Specifically, in Tables 3 and 16, we report experimental results for LLaMA-3-8B, which was released in April 2024, as detailed on https://ai.meta.com/blog/meta-llama-3/. Additionally, our experiments involve multiple model backbones, such as LLaMA-2, LLaMA-3, and OPT, thus demonstrating the versatility of our method across various LLM architectures. Furthermore, we focus on the diversity of our experimental design to demonstrate the broad applicability of our approach; the selection of models is just one part of this initiative.
>
> Regarding the models LLaMA 3.2 1B and 3B, they were released on September 25th, 2024, according to https://ai.meta.com/blog/llama-3-2-connect-2024-vision-edge-mobile-devices/. Given that the submission deadline for ICLR 2025 was October 1st, 2024, the proximity of these dates made it infeasible to include these models in our experiments. We had considered adding experiments with LLaMA-3.2-3B during the rebuttal period; however, this requires using a version of transformers that is greater than or equal to 4.43, while our current setup and implementation utilizes version 4.35.0. Given the substantial time required to adapt the code, we were unable to implement these experiments within the rebuttal phase. We believe the experiments presented in the paper adequately demonstrate the effectiveness of our methodology.

---

> > ### Author Response · Authors · 2024-11-21
> > **Response continued**
> >
> > **Response to Weakness 5:**
> >
> > First, we would like to mention that our work is concurrent with the two NeurIPS 2024 papers the reviewer referenced. However, the paper titled "Adaptive Layer Sparsity for Large Language Models via Activation Correlation Assessment (NeurIPS 2024)" has not yet been released; therefore, we are unable to access its full text or any publicly available code. Regarding "Discovering Sparsity Allocation for Layer-wise Pruning of Large Language Models (NeurIPS 2024)," we found its full text on the NeurIPS official website, but it was released after September 26th, 2024—the earliest release date we could confirm. However, we have not found any public code associated with it.
> >
> > Second, after reviewing Pruner-Zero and DSA, we have found that they focus on unstructured pruning, which differs from the structured pruning we address in our research. We wish to clarify that unstructured pruning and structured pruning for large language models are fundamentally different approaches, as they significantly differ in how they prune the model's structure. We understand Pruner-Zero and DSA as methods for finding a pruning metric and for adaptively pruning the model for each layer, respectively. Our method is orthogonal to these approaches; it can be applied with different metrics and varying pruning ratios across layers while still achieving performance gains. This is because our main contributions lie in the probing and in history-informed pruning with importance-scaled fusion.
> >
> > To demonstrate that our method is orthogonal to the approaches the reviewer mentioned, and can be applied with different metrics and varying pruning ratios across layers while still achieving performance gains, we conduct experiments on LLaMA-2-7B by applying our method to both the Wanda-sp and FLAP methods. The results are presented in the table below. Please note that **w/ PP** means we added probing and history-informed pruning with importance-scaled fusion on top of each method without modifying any part of the original framework. **PP** represents the default setting used in the main text of the paper. From the results, we can clearly see that PP generalizes well on both methods and shows significant improvements when integrated. For example, when integrated with the FLAP method under a 40\% pruning ratio, PP reduces the perplexity on WikiText2 from 38.9 to 19.9 and increases the accuracy on ARC-e from 52.5\% to 58.6\%. Similarly, when integrated with the Wanda-sp method under a 40\% pruning ratio, PP lowers the perplexity on WikiText2 from 43.8 to 19.2 and improves the accuracy on ARC-e from 54.4\% to 57.9\%.
> >
> > We hope this clarifies the distinct contributions of our work and how it complements existing methods. Furthermore, we have cited these three papers in our related work section, as they are related to unstructured pruning. Please check our revised PDF version.
> >
> > | Method           | Pruning Ratio | WikiText2 $\downarrow$ | ARC-e $\uparrow$ |
> > |------------------|---------------|------------------------|------------------|
> > | Wanda-sp w/o PP  | 20\%           | 10.6(0.1)             | 63.9(0.3)       |
> > | Wanda-sp w/ PP   | 20\%           | 8.3(0.1)              | 67.9(0.0)       |
> > | FLAP w/o PP      | 20\%           | 10.3(0.1)             | 63.1(0.1)       |
> > | FLAP w/ PP       | 20\%           | 8.3(0.1)              | 65.0(0.1)       |
> > | PP               | 20\%           | 8.1(0.1)              | 68.5(0.0)       |
> > | Wanda-sp w/o PP  | 40\%           | 43.8(1.5)             | 54.4(0.1)       |
> > | Wanda-sp w/ PP   | 40\%           | 19.2(0.2)             | 57.9(0.1)       |
> > | FLAP w/o PP      | 40\%           | 38.9(1.3)             | 52.5(0.2)       |
> > | FLAP w/ PP       | 40\%           | 19.9(0.5)             | 58.6(0.2)       |
> > | PP               | 40\%           | 16.8(0.1)             | 61.7(0.2)       |
> >
> > *Table 2: PP enhances the performance of baselines on LLaMA-2-7B.*

---

> > > ### Author Response · Authors · 2024-11-25
> > > **Thank you for upgrading your score!**
> > >
> > > Dear Reviewer ebT5,
> > >
> > > Thank you for upgrading your score! We would like to thank you again for reviewing our paper and providing valuable comments.
> > >
> > > Best regards,
> > >
> > > Authors

---

> > > > ### Comment · Reviewer_ebT5 · 2024-12-02
> > > >
> > > > Many thanks to the authors for their response. I think this paper is quite polished  and I improved my score.

---

### Official Review · Reviewer_tnLv · 2024-11-04

**Soundness:** 3
**Presentation:** 3
**Contribution:** 3
**Rating:** 6
**Confidence:** 3

**Summary:**

This paper tackles an important problem and explores saliency-aware structural pruning of neural networks. It utilizes a probing module that analyzes the L2-norm based importance for network slimming and delivers run-time inference speedup. It studies Llama2/3 and OPT LLMs and translates into performance improvements.

**Strengths:**

- The method seems to be easy to implement without incurring extensive hardware changes.
- Various ablations are provided.
- Real-time speed ups are observed on GPUs, that are a critical aspect for deployment.

**Weaknesses:**

- The method seems not to be zero-shot as it does rely on a calibration set. I could not find the detailed calibration datasets descriptions in Section 5, making it likely that the probing is looking into the final datasets to find optimal and satisfactory combinations. This makes is fairly unfair to other methods that are zero-shot on final tests given other proxy datasets. I would like to have more clarifications whether probing is looking into the final test/target metric data for numbers. If so, it's not a fair comparison to others and calibration set used for probing shall be used kept the same as other methods and report only zero-shot test values.
- Figure 3, zero probe sequence ratios all setups are below the reference line - why is it the case?
- Overall I think the method is gradient-free, but does not seem to be completely test-time per-sentence dynamic. Rather it is using batch and sequence data for a quick analysis of the saliency and deploy pruning methods.
- Table 5 indicates the method is not giving final speedups and performance boosts as strong as FLAP and Wanda-SP. This raises slight concerns for the true strength of the methods.
- Probing can incur overheads due to look aheads, and that may imply why the speedup is limited to <2x. It would be beneficial to see higher speedup ratios that may make the methods more attractive.

**Questions:**

Listed as above in the weakness.

**Details Of Ethics Concerns:**

N.A. Overall I am on the borderline for the work and would like to observe the authors response for the final judgement. My major concern is that the method utilizes probing to look into final datasets for higher numbers and such comparisons to other methods may be unfair - evidence in apple-to-apple comparisons are needed.

---

> ### Author Response · Authors · 2024-11-21
> **Response**
>
> We sincerely appreciate the reviewer's insightful feedback on the methodology of our paper. Please find our responses below:
>
> **Response to Weakness 1:**
>
> Following baseline methods in our paper, we define the "zero-shot" as a method can utilize a calibration dataset but cannot prompt the model with task-specific examples at inference time to guide its responses [1, 2]. Regarding the calibration dataset, we use the exact same subset of the C4 dataset (more specifically, c4-validation.00000-of-00008.json.gz) for all methods (including ours), in line with the baseline methods in our paper. Furthermore, the C4 dataset is not the final dataset on which we test performance. We discuss the C4 dataset in Section 5 under "Baseline" and provide detailed information in Appendix A, "Implementation Details." Furthermore, we do not prompt the model with any task-specific example during evaluation for any method (including ours).
>
> We ensure a fair comparison between the baseline methods and our method by utilizing the same calibration dataset and maintaining the same experimental settings.
>
> [1] Wei, Jason, et al. "Finetuned language models are zero-shot learners." arXiv preprint arXiv:2109.01652 (2021).
>
> [2] Brown, Tom B. "Language models are few-shot learners." arXiv preprint arXiv:2005.14165 (2020).
>
> **Response to Weakness 2:**
>
> We would like to respectfully clarify that the probe sequence ratio is not zero, but rather 0.05. In the 'Effect of Probe Combination on Performance' section, we conduct experiments to investigate how different probe sizes affect PP’s performance by varying the probe batch size from 1 to 20 (specifically, 1, 5, and 20) and the probe sequence ratio from 0.05 to 1.0 (specifically, 0.05, 0.1, 0.3, 0.5, 0.8, and 1.0). We present the experimental results in Figure 3 of the main text, demonstrating that even a small probe can significantly improve model performance. For example, when applying a small probe with a batch size of 1 and a probe sequence ratio of 0.05 on LLaMA-2-7B, the perplexity drops from 29.8 to 21.7; for OPT-13B, it decreases from 35.4 to 27.7, compared to scenarios where neither probing nor history-informed pruning of PP is applied. Note that all experiments utilize the PPsp metric for a fair comparison.
>
> **Response to Weakness 3:**
>
> We thank the reviewer for the insightful comment. While our method can be applied to per-sentence dynamic pruning, we have chosen to conduct online dynamic structured pruning in a batch-wise manner due to constraints associated with modern GPUs and the inference pipeline. Specifically, all samples within a batch need to use the same pruned weight matrix. Performing per-sentence dynamic pruning would require generating a unique pruned weight matrix for each sentence (i.e., N pruned weight matrices for N sentences), which would incur significant latency and make it impractical for real-time inference. Therefore, we conduct online dynamic structured pruning in a batch-wise manner to achieve a balance between computational efficiency and model performance.

---

> ### Author Response · Authors · 2024-11-21
> **Response continued**
>
> **Response to Weakness 4:**
>
> We would like to clarify that PP consistently outperforms FLAP and Wanda-sp across nearly all tasks, as detailed in the experimental section of the main text. In Table 5, we demonstrate that PP achieves the best Performance Runtime Ratio, which quantifies the ratio of performance degradation per unit of runtime reduction, in comparison with FLAP and Wanda-sp. In terms of performance boosts, PP's Performance Runtime Ratio values are 2.56× (95.65 compared to 37.37) and 2.85× (106.48 compared to 37.37) more efficient than those of FLAP and Wanda-sp, respectively, indicating a significantly lower rate of performance degradation compared to FLAP and Wanda-sp. With regard to the final speedups, the inference speeds of PP are on par with other structured pruning baselines, such as FLAP and Wanda-sp. It is important to note that inference speed is highly hardware-dependent and implementation-dependent. We believe that the inference speed of PP could be enhanced through further implementation optimizations or other techniques; we intend to explore these possibilities in future work.
>
> Furthermore, our method is orthogonal to the FLAP and Wanda-sp, and it can be applied with different metrics and varying pruning ratios across layers while still achieving performance gains, we conduct experiments on LLaMA-2-7B by applying our method to both the Wanda-sp and FLAP methods. The results are presented in the table below. Please note that **w/ PP** means we added probing and history-informed pruning with importance-scaled fusion on top of each method without modifying any part of the original framework. **PP** represents the default setting used in the main text of the paper. From the results, we can clearly see that PP generalizes well on both methods and shows significant improvements when integrated. For example, when integrated with the FLAP method under a 40\% pruning ratio, PP reduces the perplexity on WikiText2 from 38.9 to 19.9 and increases the accuracy on ARC-e from 52.5\% to 58.6\%. Similarly, when integrated with the Wanda-sp method under a 40\% pruning ratio, PP lowers the perplexity on WikiText2 from 43.8 to 19.2 and improves the accuracy on ARC-e from 54.4\% to 57.9\%.
>
> | Method           | Pruning Ratio | WikiText2 $\downarrow$ | ARC-e $\uparrow$ |
> |------------------|---------------|------------------------|------------------|
> | Wanda-sp w/o PP  | 20\%           | 10.6(0.1)             | 63.9(0.3)       |
> | Wanda-sp w/ PP   | 20\%           | 8.3(0.1)              | 67.9(0.0)       |
> | FLAP w/o PP      | 20\%           | 10.3(0.1)             | 63.1(0.1)       |
> | FLAP w/ PP       | 20\%           | 8.3(0.1)              | 65.0(0.1)       |
> | PP               | 20\%           | 8.1(0.1)              | 68.5(0.0)       |
> | Wanda-sp w/o PP  | 40\%           | 43.8(1.5)             | 54.4(0.1)       |
> | Wanda-sp w/ PP   | 40\%           | 19.2(0.2)             | 57.9(0.1)       |
> | FLAP w/o PP      | 40\%           | 38.9(1.3)             | 52.5(0.2)       |
> | FLAP w/ PP       | 40\%           | 19.9(0.5)             | 58.6(0.2)       |
> | PP               | 40\%           | 16.8(0.1)             | 61.7(0.2)       |
>
> *Table 1: PP enhances the performance of baselines on LLaMA-2-7B.*

---

> > ### Author Response · Authors · 2024-11-21
> > **Response continued**
> >
> > **Response to Weakness 5:**
> >
> > Probing incurs only minimal overheads in our study, as evidenced by Table 4 of the main text, where we demonstrate that probing requires approximately 1.5\% of the FLOPs compared to dense model inference. This minimal overhead contributes significantly to model performance improvements.
> >
> > To achieve higher speedup ratios, there are many unexplored areas. For example, we could generate a probe from the current batch and use it to assess the entire next layer, potentially skipping that layer entirely for the current batch. We recognize these unexplored possibilities and plan to address them in future work.
> >
> > Furthermore, we have explored the feasibility of running the probe in parallel with the actual computation of earlier pruned blocks. We present these results in Table 2. Here, **PP-Parallel** represents the approach where, when the actual computation happens on earlier pruned blocks, we generate the probe from the residuals of these earlier pruned blocks and perform the probing. **PP** represents the default setting used in the main text of the paper. The results show that we can still obtain performance gains and achieve comparable results to PP. For example, at a 40\% pruning ratio, PP-Parallel achieves a perplexity of 17.9 on WikiText2, which is close to that of PP and much lower than the 38.9 achieved by FLAP. Furthermore, PP-Parallel achieves 61.4\% accuracy on ARC-e, which is close to that of PP and much higher than the 52.5\% achieved by FLAP. However, we are just demonstrating the feasibility of further improving PP's inference speed here here; the actual parallelism is hardware-dependent and implementation-dependent.
> >
> > | Method            | Pruning Ratio | WikiText2 &darr;| ARC-e &uarr;    |
> > |-------------------|---------------|-----------|-----------|
> > | Dense             | 0\%            | 6.0(0.1)  | 67.3(0.0) |
> > | Full-Batch Probing| 20\%           | 7.3(0.1)  | 67.2(0.0) |
> > | Wanda-sp          | 20\%           | 10.6(0.1) | 63.9(0.3) |
> > | FLAP              | 20\%           | 10.3(0.1) | 63.1(0.1) |
> > | PP-Parallel       | 20\%           | 8.1(0.1)  | 67.9(0.1) |
> > | PP                | 20\%           | 8.1(0.1)  | 68.5(0.0) |
> > | Full-Batch Probing| 40\%           | 13.6(0.1) | 64.7(0.0) |
> > | Wanda-sp          | 40\%           | 43.8(1.5) | 54.4(0.1) |
> > | FLAP              | 40\%           | 38.9(1.3) | 52.5(0.2) |
> > | PP-Parallel       | 20\%           | 17.9(0.1) | 61.4(0.2) |
> > | PP                | 40\%           | 16.8(0.1) | 61.7(0.2) |
> >
> > *Table 2: Zero-shot performance of LLaMA-2-7B after pruning attention and MLP blocks without fine-tuning.*

---

> > > ### Author Response · Authors · 2024-11-25
> > > **A kind reminder**
> > >
> > > Dear Reviewer tnLV,
> > >
> > > We would like to thank you again for reviewing our paper and providing valuable comments. We believe that we have addressed your concerns. Since the Author/Reviewer Discussion Stage is ending soon and we have not heard back from you yet, we would appreciate if you could kindly let us know of any other concerns you might have, and if we can be of any further assistance in clarifying any issues.
> > >
> > > Best regards,
> > >
> > > Authors

---

> > > > ### Comment · Reviewer_tnLv · 2024-12-02
> > > > **Response**
> > > >
> > > > I would like to encourage the authors to add all the response details to the final version as this now clarifies many missing points. I have updated score to reflect this.

---

> > > > > ### Author Response · Authors · 2024-12-02
> > > > > **Thank you again!**
> > > > >
> > > > > Dear Reviewer tnLv,
> > > > >
> > > > > Thank you once again for reviewing our paper and providing valuable, positive comments. They have been extremely helpful in enhancing our work. We will incorporate all the responses into the final version of the manuscript accordingly.
> > > > >
> > > > > Best regards,
> > > > >
> > > > > Authors

---

> ### Comment · Area_Chair_mBUN · 2024-11-30
>
> Dear Reviewer,
>
> Could you kindly respond and indicate whether authors have addressed your concerns?
>
> Thanks, AC

---

### Author Response · Authors · 2024-11-21
**General response**

We appreciate all the reviewers for their valuable comments and have revised our paper accordingly. Please find the revised PDF. We conduct three experiments using different random seeds for all newly added experiments during the rebuttal period and show the standard error across these three seeds in brackets.

---

### Comment · Area_Chair_mBUN · 2024-11-28
**Reviewers please respond**

Dear Reviewers,

If you have not responded to author's rebuttal, please kindly do so as soon as possible. The deadline is Dec 2, but the authors can potentially further clarify questions if you respond earlier. Thanks!

Best,
AC

---

### Meta-Review · Area_Chair_mBUN · 2024-12-08

**Metareview:**

(a) summary: a new pruning method, Probe Pruning, for dynamically pruning LLMs. It leverages batch-wise importance and historical data to improve efficiency with minimal performance loss.

(b) strengths: novel dynamic pruning method; well motivated; easy to implement; improved efficiency; extensive experiments.

(c) weaknesses: limited comparisons; dependencies on batch sizes and sequence lengths; room for improvement in paper writing clarity.

(d) reasons for the decision: reviewers' unanimous support; a novel and practical dynamic pruning method with strong experimental results and efficiency gains.

**Additional Comments On Reviewer Discussion:**

In the rebuttal, the authors clarified comparisons, addressed batch size concerns, added experiments, and improved clarity, leading to two improved ratings.

---

### Decision · Program_Chairs · 2025-01-22

Accept (Poster)